# The Role of Astrocytes in the Central Nervous System Focused on BK Channel and Heme Oxygenase Metabolites: A Review

**DOI:** 10.3390/antiox8050121

**Published:** 2019-05-05

**Authors:** Yonghee Kim, Jinhong Park, Yoon Kyung Choi

**Affiliations:** Department of Integrative Bioscience and Biotechnology, Konkuk University, Seoul 05029, Korea; dydgml@konkuk.ac.kr (Y.K.); wls2137@konkuk.ac.kr (J.P.)

**Keywords:** astrocytes, BK channel, oxidative/nitrosative stress, heme oxygenase metabolites

## Abstract

Astrocytes outnumber neurons in the human brain, and they play a key role in numerous functions within the central nervous system (CNS), including glutamate, ion (i.e., Ca^2+^, K^+^) and water homeostasis, defense against oxidative/nitrosative stress, energy storage, mitochondria biogenesis, scar formation, tissue repair via angiogenesis and neurogenesis, and synapse modulation. After CNS injury, astrocytes communicate with surrounding neuronal and vascular systems, leading to the clearance of disease-specific protein aggregates, such as β-amyloid, and α-synuclein. The astrocytic big conductance K^+^ (BK) channel plays a role in these processes. Recently, potential therapeutic agents that target astrocytes have been tested for their potential to repair the brain. In this review, we discuss the role of the BK channel and antioxidant agents such as heme oxygenase metabolites following CNS injury. A better understanding of the cellular and molecular mechanisms of astrocytes’ functions in the healthy and diseased brains will greatly contribute to the development of therapeutic approaches following CNS injury, such as Alzheimer’s disease, Parkinson’s disease, and stroke.

## 1. Introduction

Recently, astrocytes have gained great interest because they play key roles in various functions within the central nervous system (CNS) [1]. Astrocytes have numerous thin processes that enfold blood vessels within the brain, and ensheath single as well as groups of synapses [2]. By nature of their intimate physical association with synapses, astrocytes are well-positioned to regulate extracellular concentrations of ions, neurotransmitters, and other molecules. When neurons fire action potentials, they release K^+^ ions into the extracellular space. Astrocytes express high concentrations of K^+^ channels, which act as spatial buffers. They uptake K^+^ at sites of neuronal activity (mainly synapses) and release it at distant contacts with blood vessels. K^+^ channels belong to the largest super-family of ion channels, many of which are Ca^2+^-activated [3]. On the basis of their single channel conductance, they are divided into three subfamilies: big conductance (BK), intermediate conductance, and small conductance. Astrocytic BK channels appear to be associated with voltage-gated and non-voltage-gated Ca^2+^ channels [4,5,6]. Astrocytic K^+^ and Ca^2+^ are involved in many fundamental pathophysiological functions, such as mitochondria biogenesis, cell survival, apoptosis, vascular tone, neurotransmitter release, and gene expression in the CNS [3,7,8,9,10,11,12,13]. After CNS injury, the accumulation of ions and water in astrocytes can lead to severe brain swelling.

Besides ion homeostasis, astrocytes play a key role in synapse formation and elimination [11]. Synapses underlie brain function. Once formed, synapses continue to mature and rearrange. One striking aspect of later development is the elimination of a large proportion of synapses, a process that is usually accompanied by the growth and strengthening of surviving synapses. Synapse elimination plays a key role in refining initial patterns of brain connectivity [14]. The formation and elimination of synapses involves astrocytes. Neurons form few synapses when cultured in isolation, but many more when cultured in the presence of glia [15]. Astrocytes-derived molecules, such as a large matrix protein, thrombospondin (TSP), and cholesterol, may modulate synapse function [16]. Like microglia, astrocytes also have phagocytic functions, eliminating live synapses and clearing synaptic debris [17,18].

Heme oxygenase (HO)-mediated antioxidant effects of astrocytes have been reported following CNS injury [19,20]. HO-2, a constitutively expressed form of the enzyme, regulates physiological cell function. HO-1, an inducible form of the enzyme, can be transcriptionally regulated by nuclear factor erythroid 2-related factor 2 (Nrf2) and biliverdin reductase (BVR). The Nrf2/HO-1 and/or BVR/HO-1 pathways produce carbon monoxide (CO), bilirubin (BR), and iron (Fe^2+^) by catabolizing heme. Among them, CO and BR can induce astrocytic mitochondrial biogenesis via activation of angiogenic and metabolic factors, such as AMP activated protein kinase α (AMPKα), hypoxia-inducible factor-1α (HIF-1α), estrogen-related receptor α (ERRα), and peroxisome proliferators-activated receptor γ-coactivator-1α (PGC-1α) [21,22]. Following CNS injury, astrocytes communicate with surrounding neuronal and vascular systems, leading to the clearance of disease-specific protein aggregates such as β-amyloid and α-synuclein [18,23,24]. The HO-1/CO pathway in astrocytes suppresses protein aggregate-mediated toxicity [19]. HO metabolites are involved in astrocytic ion regulation, mitochondria biogenesis, and antioxidant effects.

Bioenergetic and antioxidant defenses in the brain are coupled between neurons and astrocytes. Oxidative and nitrosative stress underlie the pathogenesis of a broad range of human diseases, in particular, neurodegenerative disorders such as Alzheimer’s disease (AD), Parkinson’s disease (PD), and stroke. Within the brain, neurons are vulnerable to excess reactive oxygen/nitrogen species (ROS/RNS); therefore, their survival relies in part on antioxidant protection promoted by astrocytes [19,23,25]. This process is partly controlled by a key glycolysis-promoting enzyme (i.e., 6-phosphofructo-2-kinase/fructose-2,6-bisphosphatase isoform 3) and is dependent on an appropriate supply of energy from astrocytes [25]. In this review, we will discuss the cellular and molecular mechanisms of astrocytes’ function in the healthy and diseased brains. Antioxidant functions of astrocytes represent promising targets for therapeutic approaches to treat CNS injury such as AD, PD, and stroke.

## 2. Astrocytes are Intimately Associated with Ion Channels

Astrocytes sense when neurons are active, by depolarizing in response to neuronal K^+^ release, and by expressing neurotransmitter receptors similar to those of neurons. For example, Bergmann glia in the cerebellum express glutamate receptors [26]. Thus, glutamate released at cerebellar synapses affects not only postsynaptic neurons, but also nearby astrocytes. The binding of these ligands to glial receptors increases the intracellular Ca^2+^ concentration. The astrocytes connect to neighboring astrocytes through gap junctions, allowing the transfer of ions and small molecules between cells [27]. Pannexin hemichannel, a vertebrate homolog of the invertebrate innexin gap junction proteins, expels adenosine triphosphate (ATP) into the extracellular space [28,29]. Ca^2+^ waves are mediated largely by the binding of ATP to the P2X7 receptor [30,31]. This spread of Ca^2+^ through the astrocyte network occur over hundreds of micrometers [27]. It is likely that this Ca^2+^ wave modulates nearby neuronal activity by triggering the release of nutrients and regulating blood flow.

Various subtypes of astrocytes have been identified according to brain region specificity in the adult mouse: fibrous and protoplasmic astrocytes (Figure 1a,b), Bergmann glia, ependymal glia, marginal glia, radial glia, perivascular glia, velate glia, and tanycytes [32]. Human astrocytes are considerably more complex when compared with rodent astrocytes. Glial fibrillary acidic protein (GFAP) expressing protoplasmic human astrocytes are larger and more complex than their rodent counterparts, and exhibit enhanced Ca^2+^ responses, increased Ca^2+^ wave velocities, and greater overlap with neighboring cells [33]. Human fibrous astrocytes are also larger in diameter when compared with those of rodents [33]. The astrocyte-to-neuron ratio in the human cortex is 1.65, while in rodents, it is approximately 0.35 [34,35]. This increased ratio of astrocytes to neurons positively correlates with brain complexity and size [35].

### 2.1. Diverse Functions of Voltage-Gated Ca^2+^ Signals in Astrocytes

Intracellular Ca^2+^ influx can be regulated in astrocytes by voltage-gated Ca^2+^ channels (VGCC) present in the cell membrane [36]. VGCC are comprised of α1-subunits, which form hetero-oligomeric complexes with modulatory accessory subunits (different β- and α2δ-subunit isoforms) [37]. Ten α1-subunit isoforms are encoded by separate genes with distinct biologic properties and exhibit different expression throughout tissue. VGCCs are divided into two groups: high-voltage-operated Ca^2+^ channels (L-, P/Q-, N-, and R-types) and low-voltage-activated channels (T-type) [38,39,40]. α1-subunits of VGCCs are as follows: L-type (Cav1.1–Cav1.4), P/Q-type (Cav2.1), N-type (Cav2.2), R-type (Cav2.3), and T type (Cav3.1–Cav3.3) Ca^2+^ channels [41].

Astrocytes ensheath small arterioles and capillaries throughout the brain, forming contacts between ends of astrocyte processes and the basal lamina surrounding endothelial cells. In adult human astrocytes, CO-mediated Ca^2+^ signaling triggers vascular endothelial growth factor (VEGF) expression and its secretion [9]. In this study, the source of Ca^2+^ was L-type VGCCs; therefore, extracellular Ca^2+^ can flow into astrocytes and activate Ca^2+^/calmodulin-dependent protein kinase kinase β (CaMKKβ)-mediated AMPKα by enhancing phosphorylation. AMPKα phosphorylation leads to the NAMPT-mediated SIRT1 activation, which results in the PGC-1α/ERRα axis and consequent VEGF expression and secretion [9]. The L-type VGCC-mediated PGC-1α/ERRα axis strongly induces human astrocytic mitochondria biogenesis [21]. Consequent increases in O_2_ consumption stabilize HIF-1α, leading to VEGF secretion and ERRα expression [22,42]. Taken together, L-type VGCC in human astrocytes may be involved in angiogenesis and mitochondria biogenesis. Deciphering the specific subunits of L-type VGCCs for the release of growth factors and mitochondria biogenesis warrants further examination.

Astrocyte hypertrophy is characterized by astrocyte swelling, enlargement, and morphological changes, and is key phenomenon in neuroinflammatory diseases [43]. The endotoxin lipopolysaccharide (LPS), high extracellular K^+^, glutamate, and ATP promote astrocyte hypertrophy in cultured astrocytes [38]. Primary cultures of mouse cortical astrocytes from newborn C57BL/6 mice show high expression of the α subunits of L-type VGCCs (i.e., Cav1.2 and Cav1.3), as well as P/Q-type VGCCs (i.e., Cav2.1) under normal physiological conditions [38]. Furthermore, LPS increases Cav1.2 expression during astrocyte hypertrophy. Astrocytes isolated from a mouse cortex transfected with small interfering RNAs (siRNAs) for Cav1.2 do not demonstrate astrocyte activation (e.g., GFAP, nestin, and vimentin expression) and proliferation (e.g., Ki67 expression) normally induced by LPS [38].

Glutamate release from active synapses can bind to G-protein-coupled receptors (GPCRs) located on astrocytes. In adult mouse hippocampal astrocytes, Gq-GPCR-linked inositol 1,4,5-triphosphate receptor type 2 (IP_3_R2)-sensitive intracellular stores can induce spontaneous and evoked Ca^2+^ release from the endoplasmic reticulum, consequently communicating with synapses and blood vessels [12,44]. Gq-GPCR-linked IP_3_R2-dependent Ca^2+^ signaling may not mediate neurovascular coupling in the visual cortex of lightly sedated responsive mice [45]. A supporting study demonstrated that small residual Ca^2+^ elevations remain in astrocytic fine processes in IP_3_R2 knockout (KO) mice [46]. Interestingly, diminished extracellular Ca^2+^ concentration markedly and reversibly abolished spontaneous signals, while IP_3_R2 KO mice also exhibited spontaneous and compartmentalized signals, suggesting that the adult mouse brain also uses extracellular Ca^2+^ for neurovascular coupling [47]. Astrocytic depolarization by high extracellular K^+^ increased the frequency of Ca^2+^ events through activation of VGCC in cultured astrocytes [48]. In this study, the morphological and structural plasticity of astrocytic processes could be translated into the frequency of spontaneous Ca^2+^ signaling [48]. Taken together, adult astrocytic Ca^2+^ can be elevated by ionic influx from extracellular space and can influence astrocytic morphological plasticity, mitochondria biogenesis, and LPS-mediated reactivity.

Inflammatory cytokines such as tumor necrosis factor α (TNFα) and prostaglandin E_2_ (PGE_2_) can be released from reactive astrocytes via kappa-light-chain-enhancer of activated B cells (NF-κB) signaling [49,50]. Reactive hippocampal astrocytes from mice, subjected to pilocarpine-induced status epilepticus, show upregulation of Cav1.3 or Cav2.1 expression [51]. Ischemic injury in the rat brain was reported to increase the expression of Cav1.2 in the reactive astrocytes located in the hippocampus and the cerebral cortex [52]. Following CNS injury, in conjunction with powerful, sequential cytosolic Ca^2+^ waves, the astrocytes promote TNFα, PGE_2_, ATP, glutamate, and D-serine secretion [53].

### 2.2. Role of the Big Conductance K^+^ (BK) Channel in Astrocytes

Presynaptic astrocytes are involved in K^+^ ion homeostasis and regulate extracellular Ca^2+^. VGCCs have spatial proximity and sometimes direct physical interaction with BK channels. A Ca^2+^ concentration of ≥10 μM is required for BK channel activation at membrane potentials of approximately 0 mV [54]. Cav channels have been known as Ca^2+^ sources for BK channel activation [55], as well as non-VGCCs such as store-operated Ca^2+^ channels, transient receptor potential family channels, and receptor-operated Ca^2+^ channels [4]. For reducing the energy consumption and fine regulation of Ca^2+^, BK and Ca^2+^ channels can associate as complexes in excitable and non-excitable cells. A recent report showed that the α2δ subunit of Cav2 or Cav1 VGCCs interacts with the N-terminus region of BK channel by preventing the association of α2δ with the calcium channel complex in dorsal root ganglion [8]. The association of α2δ with BKα reduced neuroinflammation and pain generation [8].

Levels of astrocytic endfoot Ca^2+^ may determine arteriolar dilation and constriction through BK channels in mouse brain slices [10]. Moderate elevations in astrocytic [Ca^2+^]_i_ (300–400 nM) induce arteriole dilation, while higher elevations (>700 nM) induce constriction [10]. Increases in astrocytic endfoot Ca^2+^ activate astrocytic BK channels, leading to the release of K^+^ into the perivascular space [56]. Extracellular K^+^ ions act in a concentration-dependent manner to rapidly induce both vasodilation and vasoconstriction [10]. In perivascular astrocytic endfeet of the rat hippocampus and cerebellum, BK channels also co-expressed with aquaporin 4 (AQP4) water channels, suggesting the involvement of BK channels in K^+^ redistribution and regulation of cerebral blood flow [6]. Therefore, Ca^2+^-mediated BK channel activation in astrocytes influences cerebral vascular tone and blood flow.

## 3. Astrocytes are Intimately Associated with Neuronal Functions

Astrocytes not only regulate blood flow, but also transfer mitochondria to neurons, and supply the building blocks of neurotransmitters, which fuel neuronal metabolism [2,11,57]. In addition, astrocytes can phagocytose synapses, alter neurotrophin secretion, and clear debris [14,58]. The crosstalk between astrocytes and neurons appears to begin during development; gliogenesis and synaptogenesis occur concurrently in the brain, and glial cell maturation marks the end of the synaptogenic and neuroplastic periods [59]. Besides astrocyte–neuron metabolic cooperation, astrocytic processes show little overlap with neighboring astrocytes [33]. Rather, functional networks form via gap junctions between neighboring astrocytes and contribute to highly organized anatomical domains [60].

Astrocytes become reactive in response to CNS injury. Reactive astrocytes demonstrate heterogeneity, enabling their distinction into two types (i.e., A1 and A2 reactive astrocyte), and show reversible alterations in gene expression, cell hypertrophy, and formation of a glial scar [61]. The NF-κB pathway is closely involved in the deleterious effects of A1 astrocytes on the neurovascular unit [1]. Contrarily, the scar-forming A2 reactive astrocytes are beneficial as they encapsulate the injury or seal the damaged BBB by forming a glial scar [62]. Under neuroinflammatory conditions, the A1 type astrocytes show increased expression of genes that are destructive to synapses, whereas, in ischemic conditions, the A2 type astrocytes promote the expression of the genes beneficial to neuronal survival and growth [1,62]. Activated microglia induce A1 type astrocytes by inducing IL-1α, TNFα, and C1q, and A1 astrocyte counts are elevated in AD and other neurodegenerative disorders [62]. In neuroinflammatory diseases, A1 astrocytes lose their ability to promote astrocyte–neuron connections and gain neurotoxicity.

Astrocytes upregulate and release many neurotransmitters such as glutamate, ATP, and D-serine [63], which influence paracrine signaling between astrocytes and neurons, endothelial cells, pericytes, and microglial cells. Cellular crosstalk plays a key role in the neurovascular unit under physiologic conditions, and disrupted cellular crosstalk among neuron, astrocytes, oligodendrocytes, microglia, pericytes, and endothelial cells exacerbates neurodegeneration [2,64,65].

### 3.1. Synapse Formation and Elimination by Astrocytes

Neuron–glia interactions actively control brain homeostasis through proper synaptic plasticity and neurotransmission releases (Figure 2a,b). The concept of the “tripartite synapse” refers to this cellular network involving presynaptic neurons, postsynaptic neurons, and astrocytes [11]. Astrocytes play an important role in synapse formation by constructing the tripartite synapses. A2 type astrocytes upregulate TSPs. TSPs are secreted multidomain glycoproteins found throughout the body, including in the CNS. TSP-induced synapse formation is ultrastructurally normal and shows proper alignment of the pre- and post-synaptic specializations in the tripartite synpase [16]. TSPs elicit functional presynaptic activity such as cycling of the synaptic vesicles [16]. Removal of TSPs from astrocyte-conditioned media diminishes synaptogenic activity in retinal ganglion cell cultures [16]. TSP1 and TSP2 are expressed by developing astrocytes at early postnatal stages, and their expression is reduced in adults. In addition, mice lacking both TSP1 and TSP2 have fewer excitatory synapses in the cortex, showing that astrocytes-derived TSPs may enhance excitatory (glutamatergic) synapse activity [16]. Astrocytes-derived TSPs bind to the α2δ-1 subunit of VGCCs in neurons, triggering cellular events that lead to synapse formation [66].

Astrocytes may enhance synaptic activity, resulting in changes in synaptic strength and affecting learning and memory. Long-term potentiation (LTP) relies on *N*-methyl-*D*-aspartate receptors (NMDARs). Astrocytes can activate NMDAR through Ca^2+^-dependent release of D-serine, which is an NMDAR co-agonist [67]. Clamping internal Ca^2+^ in hippocampal CA1 astrocytes in the rat blocks LTP induction at nearby excitatory synapses, and this effect can be reversed by exogenous D-serine or glycine [67]. A recent study showed that deficiency of both mouse astrocytic IP_3_R2 and IP_3_R3 significantly impaired LTP, and the application of exogenous D-serine rescued this effect [68]. Therefore, multiple subtypes of IP_3_Rs on astrocytes are concomitantly involved in D-serine secretion from astrocytes, consequently inducing NMDAR-mediated synaptic plasticity.

Phagocytic activity of astrocytes leads to synapse elimination to achieve precise neural connectivity [14]. In one study, astrocytes induced alive synapse phagocytosis via the multiple EGF-like domains 10 (MEGF10) and mer proto-oncogene tyrosine kinase (MERTK) pathways [14]. Astrocytes recognized apoptotic cells by sensing “eat-me” signals (such as phosphatidylserine) in the outer leaflet of the target’s plasma membrane, or after the target had been coated by an opsonin [17]. MEGF10-mediated astrocytic phagocytosis included the intracellular protein engulfment adaptor phosphotyrosine binding domain [14]. MERTK interacts with phosphatidylserine leading to phagocytosis [17]. Deficiency of astrocytic Megf10 and Mertk in mice leads to deficiencies in normal refinement of retinogeniculate connections in developing mice [14]. Therefore, astrocytes engulf CNS synapses to eliminate neuronal debris through MEGF10 and MERTK phagocytic pathway [14].

### 3.2. Astrocytes-Mediated Protein Aggregates Regulation

The presence of β-amyloid (Aβ) activates various signaling pathways, mainly the advanced glycation end products receptors/NF-κB pathway, which is responsible for the transcription of pro-inflammatory cytokines and chemokines in astrocytes. These cytokines include interleukin-1β (IL-1β), IL-6, inducible nitric oxide synthase (iNOS), and TNFα [23]. Astrocytes can produce Aβ, leading to oxidative stress and the production of reactive oxygen species (ROS) and reactive nitrogen species (RNS) [23]. Immature astrocytes alter gene expression in neurons, leading to the upregulated expression of C1q, a protein that initiates the complement cascade in the elimination of synapses (Figure 2c) [69].

Reactive astrocytes have dual roles in Aβ plaque degradation. The phagocytic role of reactive astrocytes in amyloid pathology may contribute to the clearance of dysfunctional synapses or synaptic debris, thereby restoring impaired neural circuits and reducing the inflammatory impact of damaged neurons [18]. In the hippocampus of amyloid precursor protein (APP)/presenilin1 (PS1) mice and AD patients, reactive astrocytes surrounding Aβ plaques enwrap and engulf axonal synapses [18]. Astrocytes also endocytose extracellular monomeric and oligomeric Aβ through actin polymerization [70]. In contrast, reactive astrocytes-mediated γ-aminobutyric acid (GABA) release exacerbates impaired spike probability, synaptic plasticity, and cognitive functions in APP/PS1 mice [71]. Brains from postmortem AD patients show increased levels of GABA and monoamine oxidase B, predominantly in reactive astrocytes [71]. When astrocytes engulf large amounts of Aβ protofibrils in transgenic (Tg)-ArcSwe mice (harboring the human Arctic and Swedish APP mutations), incomplete digestion results in a high intracellular load of toxic partially truncated Aβ, and severe lysosomal dysfunction [72]. Microvesicles containing N-terminally truncated Aβ from astrocytes induce cortical neuronal apoptosis [72].

Tau is a microtubule-associated protein (MAP) expressed in axons. Modified tau proteins undergoing abnormal polymerization form the characteristic lesion called the neurofibrillary tangle [73]. Progressive neurodegenerative disorders with pathological findings of filamentous inclusion bodies composed of MAP tau are referred to as tauopathies and, moreover, tau accumulation in astrocytes has been investigated [74]. Forman et al. demonstrated the role of astrocytes in tau pathology by developing a Tg mouse expressing the T34 human tau isoform under the control of the astrocyte-specific GFAP promoter [75]. Expression of human tau specifically in astrocytes leads to deleterious consequences within astrocytes and non-cell autonomous effects on the neurovascular unit. Aged GFAP/tau Tg mice (>20 months) demonstrated mild breakdown of the blood–brain barrier (BBB) and axonal degeneration with disruption of the associated myelin [75], suggesting that tau pathology in astrocytes is involved with neurovascular miscommunication. Another study showed that blocking the IL-1 receptor (IL-1R) alleviated cognitive deficit, partly reduced fibrillar Aβ formation, significantly attenuated tauopathies, and restored neuronal β-catenin pathway in the 3xTg-AD mouse model of AD (a triple-transgenic model with three mutant human genes: APP_swe_**,** PS1_M146V_, and tau_P301L_) [76]. Inhibition of IL-1R in 3xTg-AD mice by intraperitoneal injection of 200 μg of IL-1R antibody reduced the astrocyte-derived Ca^2+^ binding protein, S100β [76]. Astrocyte-derived S100β can upregulate iNOS in cultured rat astrocytes through NF-κB activation [77], exacerbating oxidative/nitrosative stress in an autocrine manner. Astrocytic S100β secretion leads to tau hyperphosphorylation by inducing glycogen synthase kinase-3β phosphorylation-mediated β-catenin degradation in human neural stem cells in a paracrine manner [78]. Therefore, healthy astrocytes contribute to proper neuronal functions and reactive inflammatory astrocytes accelerate neuronal pathology.

### 3.3. Mitochondria Functions in Astrocyte–Neuron Crosstalk

The accumulation of damaged mitochondria is a hallmark of aging and age-related neurodegeneration, including AD, PD, and stroke. Intercellular communication between astrocytes and neurons involves organelle exchange, including the unidirectional or bidirectional transfer of healthy mitochondria. Mitochondria contain their own circular genome encoding selected subunits of the oxidative phosphorylation complexes. Recent findings revealed that mitochondria can traverse cell boundaries and be transferred between cells [79]. Astrocytic mitochondria may regulate Ca^2+^-mediated signaling, apoptosis, and cell metabolism. In addition, intercellular mitochondrial transfer rescues damaged cells by restoring aerobic respiration from mitochondrial dysfunctions related to ischemic stress. Some stress conditions such as hyperglycemic or low-serum medium, as well as cytokines, stimulate the transfer of vesicles, proteins, and mitochondria [79,80]. Astrocytic mitochondria can be released and transferred to neuron for neuronal survival and functional recovery from ischemic stroke in the mouse [57]. Astrocytes-derived extracellular vesicles can be one of the methods of mitochondrial transfer [57]. Retinal ganglion cell axons also transfer mitochondria to neighboring astrocytes for degradation [81]. Astrocytes processes contain mitochondria, which are decreased by in vitro oxygen/glucose deprivation (OGD) followed by reperfusion injury [82]. This insult results in delayed fragmentation and mitophagy (the selective degradation of mitochondria by autophagy), which often occurs following damage or oxidative stress in the mitochondria. Excessive glutamate uptake in astrocytes can trigger mitochondria loss and dramatic increases in Ca^2+^ signaling in astrocytic processes [82].

Axonal protrusions and evulsions within the optic nerve head contain mitochondria that are derived from neurons and are degraded by lysosomes of healthy astrocytes [81]. A recent study demonstrates the beneficial effects of mitophagy on Aβ degradation in an APP/PS1 mouse model [83]. In addition, enhancing mitophagy reduces tau hyperphosphorylation in human neuronal cells, leading to improved memory [83]. Therefore, the transfer of defective neuronal mitochondria into astrocytes where they may undergo mitophagy represents a potential therapeutic intervention.

### 3.4. Effects of Astrocytic HIF-1α on Neurovascular Functions

HIF-1α, a master regulator of the cellular response to hypoxia, is a transcription factor that is stabilized and activated by hypoxia [84]. Astrocytic HIF-1α may be detrimental or beneficial for neurovascular survival. Cerebral ischemia leads to hypoxia, oxidative stress, and inflammation in the acute phase. HIF-1α may exacerbate the inflammatory response through activation of glial T-cell immunoglobulin and mucin domain protein (TIM)-3 [85]. Astrocytic TIM-3 is upregulated in the hypoxic penumbra in the ischemic brain, and ischemic injury-mediated damage is reduced in mice following treatment with TIM-3-neutralizing antibody or HIF-1α KO brain [85]. Administration of a lentiviral vector expressing TIM-3 in HIF-1α-deficient mice significantly increases infarct size and neurological deficit [85]. Part of A-kinase anchor protein 12 (AKAP12) protein is expressed in retinal astrocytes, and can downregulate HIF-1α protein stability in an O_2_-dependent manner [86]. AKAP12-overexpressing hypoxic astrocytes lead to increased tight junction proteins and reduction of a permeability factor, vascular endothelial growth factor (VEGF) in retina microvascular endothelial cells [86,87], suggesting that oxidative stress-mediated HIF-1α stability and secretion of VEGF in hypoxic astrocytes may be deleterious for the BBB.

Activation of HIF-1α under ischemic conditions has been shown to protect astrocytes against oxidative stress. In cultured rat astrocytes, HIF-1α has a protective role by promoting astrocyte survival after excitotoxic and hypoxic injury [88,89]. A HIF-1α downstream factor, VEGF, also has protective effects against ischemic injury by stimulating angiogenesis, mitochondria biogenesis, neuronal survival, and neurogenesis [90,91,92,93]. Even though VEGF can be deleterious for the BBB in acute phase, VEGF is involved in regeneration in chronic phase of CNS injury. Aβ-dependent astrocyte activation can result in reduction of HIF-1α stability and glycolysis in chronic phase [94]. Aβ increases the production of ROS through the activation of nicotinamide adenine dinucleotide phosphate oxidase (NADPH oxidase, NOX) and consequently reduces glycolysis [94]. Even though consistent HO-1 overexpression may lead to accumulation of iron in astrocytes [95], transient exposure of CO and BR possesses significant antioxidant effects [43]. Considering that recovery of astrocytic metabolic integrity in chronic oxidative stress conditions may be pivotal for neurovascular demands, HIF-1α-related signals may be critical in the chronic phase of oxidative stress.

### 3.5. Astrocytes and Oxidative/Nitrosative Stress

Non-reactive astrocytes become biochemically altered, and referred to as reactive, after oxidative/nitrosative injuries. Reactive astrocytes profiling can be performed on mice treated with a systemic injection of LPS to induce neuroinflammation, or subjected to middle cerebral artery occlusion (MCAO) to induce cerebral ischemia [96]. Hypoxia or inflammation promotes reactive astrocytes. Reactive astrocytes have dual role in axonal death and regeneration. TNFα-mediated induction of the NF-κB pathway stimulates the increase of the proinflammatory enzyme, cyclooxygenase-2 and, additionally, increases tau hyperphosphorylation via c-Jun N-terminal kinase [97]. A2 reactive astrocytes are beneficial as they promote neurotrophic factors partly through the Janus kinase-signal transducer and activator of transcription protein 3 pathway [62].

Oxidative/nitrosative stress may be caused by overproduction of ROS/RNS and impairment of the cellular antioxidant defense capacity [98]. Astrocytes have robust antioxidant abilities, which are much more limited in neurons [99]. Astrocytes are significantly more sensitive to 1-methyl-4-phenylpyridinium (MPP^+^)- or α-synuclein-mediated HO-1 induction than neurons [99]. In one study, a HO-1 inducer, cobalt protoporphyrin IX (CoPPIX), protected astrocytes from MPP^+^- or α-synuclein-mediated mitochondria membrane collapse. Nrf2 deficiency increases 1-methyl-4-phenyl-1,2,3,6-tetrahydropyridine (MPTP)-induced PD-like lesions in mice, which are alleviated by GFAP-Nrf2 Tg mice [100]. This interesting result indicates that Nrf2 expression restricted to astrocytes may be sufficient to protect against MPTP.

## 4. Astrocytes and Diseases

Neurodegenerative disorders exhibit dysregulation of Ca^2+^ homeostasis as a result of excitotoxicity, perturbed energy metabolism and oxidative stress, and compromised cellular Ca^2+^-regulating systems. In stressed astrocytes, extracellular Ca^2+^ enters the cell and elevates cytosolic and mitochondrial Ca^2+^ concentrations. Excessive Ca^2+^-mediated signaling may accelerate ROS/RNS production in astrocytes. The antioxidant capacity of astrocytes is important for reducing ROS/RNS production and inducing cellular protection. Beneficial and/or deleterious functions of reactive astrocytes in neurodegenerative diseases such as AD, PD, and stroke will be discussed in the following section.

### 4.1. Alzheimer’s Disease

AD is a devastating neurodegenerative CNS injury showing toxic protein aggregates such as Aβ plaques and tau tangles. A pathological increase in the amount of Aβ can induce astrocytic Ca^2+^ ion disturbances [23]. In addition, Aβ-induced disruption of NMDAR or Ca^2+^ channels in astrocytes and neurons may be related to abnormal Ca^2+^ signaling and Ca^2+^-dependent release of D-serine, breaking down neuron-glia and glia-glia networks. As a result, Ca^2+^-mediated cAMP response element binding protein phosphorylation and brain-derived neurotrophic factor expression in neuron can be modified by reactive astrocytes, leading to impaired synaptic plasticity and altered neurotransmitter release.

BK channels of astrocytic endfeet are activated by elevation in astrocytic Ca^2+^ concentration [56]. Altered Ca^2+^ influx by Aβ may downregulate BK expression in brains from mouse models of vascular amyloid deposition representing stages of cerebral amyloid angiopathy, consistent with general disruption of the neurovascular unit in AD [101]. Reduced BK activity results in indirectly enhanced Ca^2+^ influx [102] and consequent Ca^2+^ overload-mediated neuronal and glial death in AD. In 3xTg-AD mice, both ventricular injection of isopimaric acid, a BK channel opener, and use of transcranial magnetic stimulation (TMS) ameliorate cognitive deficits and cortical hyperexcitability [103]. BK channel activity can be recovered by TMS or TMS-frequency-dependent scaffold protein Homer1a expression, which in turn enhances LTP [103]. In addition, BK channel activation is connected with reduction in brain Aβ content, showing the beneficial circuit of BK activation and Aβ reduction in an AD mouse model [103].

Astrocytes participate in the degradation and removal of Aβ as they express various proteases involved in the enzymatic cleavage of Aβ. Peptidases such as neprilysin (NEP), insulin-degrading enzyme (IDE), endothelin-converting enzyme (ECE), and matrix metalloproteinases (MMPs) are expressed in astrocytes, and are involved in the degradation of extracellular forms of Aβ [104,105]. NEP, a 90–110 kDa plasma membrane glycoprotein, is the member of the zinc metallopeptidase family and is reduced in AD brains [106]. IDE and ECE are upregulated, showing a positive correlation with brain Aβ levels [107].

Glucose transporter-1, a downstream gene of HIF-1α [87], at the BBB decreases in individuals with AD [108], suggesting that astrocytes have a shortage of metabolic substrates. If putative therapeutic drugs can boost astrocyte metabolic pathways, it may be possible for healthy astrocytes to contribute to vascular dilation via BK-AQP4 channel activation and improve neuronal energy metabolism via mitochondria transfer in age-mediated neurodegenerative diseases.

### 4.2. Parkinson’s Disease

The neuropathology of PD is characterized by the selective death of dopaminergic neurons in the substantia nigra. Protein aggregations, termed Lewy bodies, have been reported in PD. α-Synuclein is composed of 140 amino acids expressed dominantly at pre-synaptic terminals. In physiological conditions, astrocytes also express α-synuclein at very low levels. α-synuclein may contribute to synaptic vesicle biogenesis and dynamics, and neurotransmission [109]. In pathological conditions, accumulation of α-synuclein aggregates in intracellular and extracellular dopaminergic neurons are one of the main hallmarks of PD. Astrocytes can internalize α-synuclein from the extracellular space via endocytosis-mediated degradation in lysosomes [24]; however, at high α-synuclein concentrations, this process may become overloaded, resulting in the accumulation of α-synuclein in the cytosol and the formation of Lewy bodies.

Glial cells such as astrocytes and microglia have the ability to uptake and clear misfolded/aggregated extracellular α-synuclein [24]. Although neurons also internalize α-synuclein, astrocytes appear to be more efficient scavengers. Endocytosed α-synuclein has been shown to localize to the lysosome, in conformation-sensitive, cell- and receptor-type specific processes [110]. Astrocytic phagocytosis may be involved in α-synuclein degradation and clearance [17].

The loss of dopaminergic neurons can affect astrocytic Ca^2+^ homeostasis. Astrocytic Ca^2+^ signals appear to be regulated by dopamine. Dopamine elevates or lowers astrocytic Ca^2+^ depending on the receptors involved [111]. Dopaminergic neurons also regulate muscle movement, requiring large amounts of energy provided by mitochondria. PTEN-induced putative kinase 1 (PINK1) has biologic functions involved in mitophagy. PINK1 expression is Ca^2+^-dependent, and it protects against overloaded Ca^2+^ excitability by regulating mitochondrial Ca^2+^ transport [112,113]. PINK1 loss is associated with oxidative damage, mitochondrial Ca^2+^ mishandling, mitochondrial dysfunction, and increased neuronal vulnerability [114]. PINK1 also has an important role in astrocytes development. Brains from *Pink1*-KO mice exhibit a reduction in the number of astrocytes compared with wild-type (WT) mice [115]. PINK1 expression is higher in human astrocytes compared with neurons [116]. Impaired responses of antioxidant proteins such as HO-1 and superoxide dismutase 2 are related to human PINK1 gene mutation and accelerated oxidative stress and neurodegeneration [117]. Astrocyte-specific human *HO-1* gene over-expression exhibit increases in *PINK1* gene expression in astrocytes, but not neurons in the substantia nigra of Tg mice [95]. Therefore, astrocyte proliferation capacity and overall number may be associated with astrocytic antioxidant functions via PINK1.

### 4.3. Stroke

Cerebral ischemia/reperfusion injury is followed by a delayed secondary pathology including excitotoxic and inflammatory responses. BKα KO mice display a larger infarct volume, more severe neurological scores, and higher mortality than their WT littermates following ischemia/reperfusion injury [118]. Administration of a BK channel opener (BMS-204352) was intravenously injected after MCAO in rats, and showed reduced cortical infarct size [5]. Despite promising preclinical results, the therapeutic efficacy of BMS-204352 failed to demonstrate improvement in a phase III clinical trial involving acute stroke patients (Reviewed in the work of [3]). Deciphering the specific cell-type (i.e., astrocyte, neuron) that benefit from activation of BK channels to lead to neuroprotection requires further examination.

Astrocytes are much more resilient to ischemic/reperfusion-mediated inflammatory injury than neurons and may play an important role in the development of damage. Cerebral damage is a result of O_2_ and energy depletion, as well as subsequent acidosis, inflammation, glutamate excitotoxicity, and ROS/RNS generation [119]. In this state, reactive astrocytes exert biphasic functions, that is, beneficial or detrimental depending on the regulating factors, metabolic conditions, microenvironment for O_2_ supply, and ROS/RNS modulation. Transient OGD causes delayed fragmentation and autophagic degradation of mitochondria through excessive Ca^2+^ influx in rat astrocytic processes [82]. The production of functional mitochondria from astrocytes can affect adjacent ischemic/reperfusion affected neurons, consequently enhancing neuronal survival and improving functional outcome [57].

## 5. Therapeutic Effects of HO Metabolites in CNS Injury

Pharmacological interventions specifically targeting only neurons are unlikely to succeed, because it is not feasible to preserve neuronal viability in an environment that fails to meet essential metabolic requirements. An emerging concept for CNS repair is to target healthy astrocytes, which may contribute to improved cellular communication with microvascular and thousands of synapses. Further, astrocytes may play a role in diminishing inflammatory responses, reducing protein aggregates and enhancing mitochondria transfer, all of which likely contribute to repair following CNS injury. Meanwhile, other glia cells (i.e., oligodendrocytes and microglia) may further improve recovery.

HO metabolites such as CO and BR may increase these effects, leading to regeneration of vascular and neuronal systems [43]. HO-1 inducers (e.g., CoPPIX, CORM) have more beneficial effects on the survival of astrocytes and neurons [65,99,120] compared with HO-1 overexpression in astrocytes [95] following CNS injury. Persistent expression of HO-1 in astrocytes is deleterious, as excessive accumulation of iron can lead to inflammation and cell death [95]. HO-1 inducer transiently upregulates HO-1, concomitantly with enhanced levels of antioxidant proteins such as Nrf2 or BVR [121,122], as well as mitochondrial ferritin [99]. More investigation is needed to elucidate the underlying mechanisms involved. It is possible that HO-1 inducers in astrocytes have more efficient iron buffering systems and antioxidant effects than HO-1 overexpression in astrocytes.

### 5.1. Carbon Monoxide

Astrocytes-derived CO production has been reported to contribute to vasodilation [123], leading to the supply of O_2_ and nutrients to neighboring cells. Adenosine diphosphate (ADP) and NO are important signaling molecules in the brain, and both ADP and NO donors increase pial arteriolar diameter [124]. Dilation in response to ADP and ADP-dependent CO production were blocked by the metal porphyrin inhibitor of HO in astrocytes and cerebral microvessels [124]. CO and NO can activate BK channels in endothelial cells [125]. In addition, astrocytic-derived CO activates BK channels in smooth muscle cells directly, as well as via an NO-dependent pathway [126]. Therefore, astrocyte-derived CO can diffuse into endothelial and smooth muscle cells, leading to BK channel activation and consequent vasodilation.

CO can reduce ROS/RNS production following CNS injury [65,120,127], likely inhibiting reactive astrocyte-mediated neuroinflammatory effects. The inflammatory cytokine TNFα can produce CO, a gaseous antioxidant neurotransmitter in cerebral microvascular endothelial cells [128]. CORM-A1, a form of CO-releasing molecule, inhibits NOX4-mediated ROS generation and enhances cytoprotection [128]. Therefore, there may be a feedback control of NOX-4-mediated ROS and CO production, wherein NOX4 activity can be inhibited by CO, preventing excessive ROS generation and oxidative stress (Figure 3).

Pretreatment of human astrocytes cells with CORM-2 can induce HIF-1α protein stability via downregulation of PHD2-mediated proteasomal degradation by enhanced O_2_ consumption (hypoxia) rather than by ROS generation [22]. In contrast, Aβ increases the long-term production of ROS through NOX and reduces the amount of HIF-1α via the action of the proteasome in rat astrocytes [94]. Considering that HIF-1α regulates astrocytic metabolism and glycolysis as well as the prevention of glial activation [94], CO-mediated HIF-1α stabilization in astrocytes may have therapeutic potential as an intervention for energy-demanding CNS injuries. In addition, CORM-2 protects rat astrocytes and hippocampal neurons against Aβ toxicity in a concentration-dependent and HO-1-dependent manner [19,129]. CO suppresses NOX-derived ROS production caused by Aβ in rat astrocytes [19]. Crosstalk between NF-κB and ROS activates A1 type astrocytes [130]. Therefore, CO may downregulate the ROS/NF-κB circuit in astrocytes, consequently reducing transformation of inflammatory A1 astrocytes after CNS injury (Figure 3). Further investigation is necessary to confirm the effects of CO on the transition of astrocytes to A1 type.

CO prevents apoptosis in rat astrocytes by inducing Bcl-2 expression, which enhanced ATP generation, cytochrome c oxidase enzymatic activity, and mitochondrial oxidative phosphorylation [131]. CO also stimulates PGC-1α-mediated astrocytic mitochondria biogenesis [21]. Therefore, properly regulating the concentration of CO could possess therapeutic potential via astrocytes modulation.

### 5.2. BR

The BVR/HO-1 pathway may help restore a more favorable tissue redox condition by promoting antioxidant BR production [95]. BVR KO mice have a substantially increased susceptibility to a broad range of chemical toxicities and AD conditions associated with oxidative pathology [132,133]. In cultured human astrocytes cells, various factors such as HIF-1α, PGC-1α, ERRα, and VEGF can be synergistically upregulated by the combinatory treatment of CO and BR through extracellular Ca^2+^ signaling [9,21,22]. These factors are involved in regenerative effects through mitochondria biogenesis, angiogenesis, and neurogenesis [93,134,135,136].

## 6. Conclusions

Antioxidants have been applied in clinical trials because they are neuroprotective in animal models. Injection of oxidative/nitrosative stressed mice with phenyl-α-tert-butyl nitrone, a ROS/RNS scavenger, did not result in a significant improvement of behavioral function [65]. Phase III clinical trials testing a combination of an antioxidant with a drug (i.e., vitamin E plus memantine or selenium) for prevention or treatment of AD have failed (reviewed in the work of [137]). Despite negative outcomes in large clinical trials, antioxidant agents are still important factors that may influence the critical balance between production and elimination of ROS/RNS.

HO-1 inducers including CO reduce ROS/RNS production, and rescue motor and cognitive functions following traumatic brain injury or ischemia/reperfusion injury [65,120]. We assume that antioxidant-treated astrocytes may accelerate improved cellular communication with microvascular and thousands of synapses after CNS injury [66,138]. Diminished inflammatory responses, reduced protein aggregates, and enhanced metabolic capacity from healthy astrocytes can contribute to repair of CNS injury.

## Figures and Tables

**Figure 1 antioxidants-08-00121-f001:**
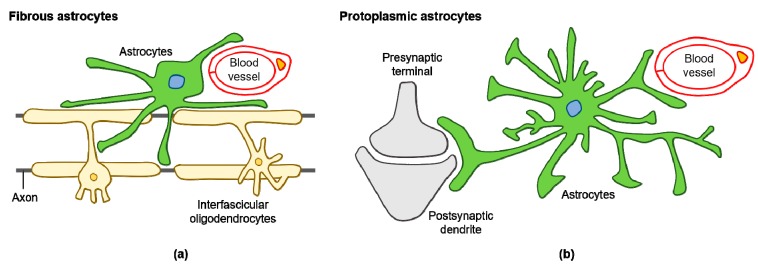
Diagram of fibrous and protoplasmic astrocytes in the central nervous system (CNS): (**a**) fibrous astrocytes in the white matter contact with blood capillaries and axons; (**b**) protoplasmic astrocytes in the gray matter contact with blood capillaries and synapses.

**Figure 2 antioxidants-08-00121-f002:**
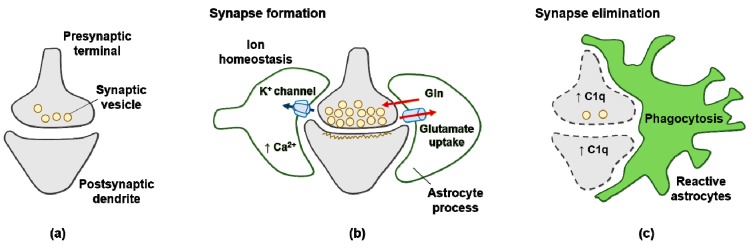
Signals from astrocytes promote synapse formation and elimination: (**a**) neurons without astrocytes form less synaptic neurotransmitter vesicle; (**b**) astrocytes promote the maturation of both pre- and postsynaptic elements of the synapse; (**c**) reactive astrocytes upregulate expression of the complement component C1q in neurons, leading to elimination by astrocyte-mediated phagocytosis.

**Figure 3 antioxidants-08-00121-f003:**
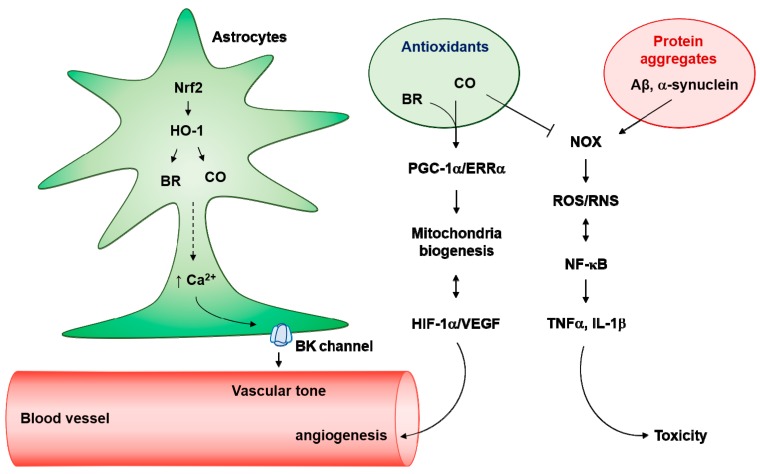
Antioxidant effects of astrocytes on neurotoxic protein aggregates. Nuclear factor erythroid 2-related factor 2 (Nrf2)/heme oxygenase (HO-1) pathway produces carbon monoxide (CO) and bilirubin (BR) in astrocytes, which induces Ca^2+^-enhanced K^+^ channel activation. Astrocytes endfeet big conductance K^+^ (BK) channels can regulate vascular tone. In addition, the Ca^2+^-mediated peroxisome proliferators-activated receptor γ-coactivator-1α (PGC-1α)/estrogen-related receptor α (ERRα) axis can facilitate mitochondria biogenesis. Elevated O_2_ consumption by mitochondria biogenesis results in hypoxia-inducible factor-1α (HIF-1α) stability and vascular endothelial growth factor (VEGF)-mediated angiogenesis. Interestingly, activated HIF-1α/VEGF may induce astrocytic mitochondrial biogenesis. In pathologic conditions, neurotoxic protein aggregates such as Aβ and α-synuclein can facilitate nicotinamide adenine dinucleotide phosphate oxidase (NADPH oxidase, NOX)-mediated reactive oxygen species (ROS) production and kappa-light-chain-enhancer of activated B cells (NF-κB) activation. NF-κB-mediated inducible nitric oxide synthase (iNOS) activation results in the production of and reactive nitrogen species (RNS). Crosstalk between ROS/RNS and NF-κB stimulates the secretion of neurotoxic cytokines such as tumor necrosis factor α (TNFα) and interleukin (IL)-1β. Antioxidants (i.e., CO and BR) may block the NOX-mediated ROS/RNS generation in astrocytes.

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
