# Peer review of "The Role of Astrocytes in the Central Nervous System Focused on BK Channel and Heme Oxygenase Metabolites: A Review"

_antioxidants, 2019, doi:10.3390/antiox8050121_

Round 1

Reviewer 1 Report

The authors have submitted an interesting review regarding the role of astrocytes in numerous functions of the healthy and damaged CNS and some of the cellular and molecular mechanisms that are involved. They also discuss the potential of antioxidants-treated astrocytes as therapeutic agents for CNS diseases.

The manuscript is well written, clearly organized, and the bibliography is quite actual. I only suggest to describe if A1 and A2 astrocytes are fibrous or protoplasmic, respectively.

Author Response

Response to Comments by Reviewer 1

The authors have submitted an interesting review regarding the role of astrocytes in numerous functions of the healthy and damaged CNS and some of the cellular and molecular mechanisms that are involved. They also discuss the potential of antioxidants-treated astrocytes .as therapeutic agents for CNS diseases.

Comment #1 : The manuscript is well written, clearly organized, and the bibliography is quite actual. I only suggest to describe if A1 and A2 astrocytes are fibrous or protoplasmic, respectively.

Response to comment #1:

Astrocytes present phenotypic heterogeneity, and may be classified as fibrous and protoplasmic astrocytes1. A1 and A2 astrocytes are types of “reactive astrocytes,” converted from diverse astrocytes, and were so termed by Liddelow S. A. et al.2. We have revised the original description (original lines 243-249 and 354-359) and included a more detailed description of the A1 and A2 astrocytes in (revised lines 185-195) the revised manuscript.

(Lines 185-196) Astrocytes become reactive in response to CNS injury. Reactive astrocytes demonstrate heterogeneity, enabling their distinction into two types (i.e., A1 and A2 reactive astrocyte), and show reversible alterations in gene expression, cell hypertrophy, and formation of a glial scar3. The NF-κB pathway is closely involved in the deleterious effects of A1 astrocytes on the neurovascular unit4. Contrarily, the scar-forming A2 reactive astrocytes are beneficial as they encapsulate the injury or seal the damaged BBB by forming a glial scar2. Under neuroinflammatory conditions, the A1 type astrocytes show increased expression of genes that are destructive to synapses, whereas, in ischemic conditions, the A2 type astrocytes promote the expression of the genes beneficial to neuronal survival and growth2, 4. Activated microglia induce A1 type astrocytes by inducing IL-1a, TNFa, and C1q, and A1 astrocyte counts are elevated in AD and other neurodegenerative disorders2. In neuroinflammatory diseases, A1 astrocytes lose their ability to promote astrocyte-neuron connections and gain neurotoxicity.

(Original Lines 243-249) Under neuroinflammatory conditions, A1 type astrocytes show enhanced expression of genes that are destructive to synapses, while in ischemic conditions A2 type astrocytes promote expression of genes that are beneficial to neuronal survival and growth2, 4. Activated microglia induce A1 type of astrocytes by secretion of IL-1a, TNFa, and C1q, and A1 astrocytes are upregulated in AD and other neurodegenerative disorders2. In neuroinflammatory diseases, A1 astrocytes lose their ability to promote astrocyte-neuron connections and gain neurotoxicity.

(Original Lines 354-359) Reactive astrocytes have dual role in axonal death and regeneration. The NF-κB pathway is closely related to the deleterious A1 astrocyte-mediated effects on the neurovascular unit4TNFa-mediated induction of the NF-κB pathway stimulates the increase of the proinflammatory enzyme, cyclooxygenase-2, and additionally, increases tau hyperphosphorylation via c-Jun N-terminal kinase (JNK)5. Scar-forming A2 reactive astrocytes are beneficial as they encapsulate injury or seal off the damaged BBB by forming a glial scar. The JAK-STAT3 pathway is involved in A2 astrocytes activation2. A2 reactive astrocytes are beneficial as they promote neurotrophic factors, partly through the Janus kinase-Signal transducer and activator of transcription protein 3 JAK-STAT3 pathway2.

References

1.              Sosunov, A. A.; Wu, X.; Tsankova, N. M.; Guilfoyle, E.; McKhann, G. M., 2nd; Goldman, J. E., Phenotypic heterogeneity and plasticity of isocortical and hippocampal astrocytes in the human brain. J Neurosci 2014, 34 (6), 2285-98.

2.              Liddelow, S. A.; Guttenplan, K. A.; Clarke, L. E.; Bennett, F. C.; Bohlen, C. J.; Schirmer, L.; Bennett, M. L.; Munch, A. E.; Chung, W. S.; Peterson, T. C.; Wilton, D. K.; Frouin, A.; Napier, B. A.; Panicker, N.; Kumar, M.; Buckwalter, M. S.; Rowitch, D. H.; Dawson, V. L.; Dawson, T. M.; Stevens, B.; Barres, B. A., Neurotoxic reactive astrocytes are induced by activated microglia. Nature 2017, 541 (7638), 481-487.

3.              Anderson, M. A.; Ao, Y.; Sofroniew, M. V., Heterogeneity of reactive astrocytes. Neurosci Lett 2014, 565, 23-9.

4.              Liddelow, S. A.; Barres, B. A., Reactive Astrocytes: Production, Function, and Therapeutic Potential. Immunity 2017, 46 (6), 957-967.

5.              Ploia, C.; Antoniou, X.; Sclip, A.; Grande, V.; Cardinetti, D.; Colombo, A.; Canu, N.; Benussi, L.; Ghidoni, R.; Forloni, G.; Borsello, T., JNK plays a key role in tau hyperphosphorylation in Alzheimer's disease models. J Alzheimers Dis 2011, 26 (2), 315-29.

Reviewer 2 Report

Astrocytes have many important roles in the brain and yet have not received the attention that they deserve. Thus, this review has the potential to bring more attention to the diverse roles of astrocytes in the maintenance of proper brain function. However, in its current form, it is very repetitive and also very difficult to follow as the logic of many attempted connections between different sets of results is lacking. The review also needs to be reorganized so that specific topics are discussed only once, not repeatedly under different subtitles. For example, AD is discussed under protein aggregation and then again under AD. Examples of specific problems and concerns are listed below but this is by no means a complete list. A thorough re-evaluation and re-write is needed.

1. Introduction: this is much too long and detailed. It should just explain the overall topic and then highlight what will be covered in the rest of the review.

2. line 66: what key glycolysis promoting enzyme?

3. lines 81-85: There is no logical connection between the second sentence and the rest of the paragraph.

4. lines 87-95: References are needed.

5. lines 131-139: There is no logical connection between the first sentence and the rest of the paragraph.

6. line 156: COX2 is not a cytokine.

7. lines 156-162: The first and second sentences are not well connected nor are they at all connected to the last two sentences.

8. line 165: Define BK channels.

9. lines 194-195: Earlier many of the same compounds described here as neurotransmitters were described as toxic cytokines. Which is it?

10. lines 200-212: What does TSP mean? It is not clear what role TSPs have in the tripartite synapse.

11. lines 222-235: This paragraph describes two different processes that appear to have no clear relationship. Why are they combined?

12. lines 244-245: what are A1 and A2 type astrocytes?

13. lines 269-271: tau is not a low molecular weight protein (~56 kD) and a tauopathy is a disease caused by aggregation of tau. The aggregates are called neurofibrillary tangles not tauopathies.

14. lines 277-284: Not clear how this treatment is beneficial or directly relevant to astrocytes.

15. section 3.3: Much of this is discussed elsewhere. No need to repeat.

16. section 3.4: Much of this is discussed earlier. Should consolidate and not repeat.

17. lines 347-350: Relevance of these sentences to the rest of the paragraph is not clear nor is the relevance to the review in general.

18: lines 443-452: Not clear why this is relevant to astrocytes.

19: lines 475-476: Why would there be a difference? Why would induction of HO-1 by chemicals provide better buffering than HO-1 overexpression? The authors need to explain.

20: lines 478-486: Much of this was covered earlier. No need to repeat.

21: lines 532-538: The authors need to explain what was being tested here. Clinical trials of what?

Author Response

Response to Comments by Reviewer 2

Astrocytes have many important roles in the brain and yet have not received the attention that they deserve. Thus, this review has the potential to bring more attention to the diverse roles of astrocytes in the maintenance of proper brain function. However, in its current form, it is very repetitive and also very difficult to follow as the logic of many attempted connections between different sets of results is lacking. The review also needs to be reorganized so that specific topics are discussed only once, not repeatedly under different subtitles. For example, AD is discussed under protein aggregation and then again under AD. Examples of specific problems and concerns are listed below but this is by no means a complete list. A thorough re-evaluation and re-write is needed.

Comment #1: Introduction: this is much too long and detailed. It should just explain the overall topic and then highlight what will be covered in the rest of the review.

Response to comment #1: Per your suggestion, we have reduced the word count and reorganized the introduction section.

(Lines 25-74) 1. Introduction

Recently, astrocytes have gained great interest because they play key roles in various functions within the central nervous system (CNS)1. Astrocytes have numerous thin processes that enfold blood vessels within the brain, and ensheath single and groups of synapses2. By nature of their intimate physical association with synapses, astrocytes are well-positioned to regulate extracellular concentrations of ions, neurotransmitters, and other molecules. When neurons fire action potentials, they release K+ ions into the extracellular space. Astrocytes express high concentrations of K+ channels, which act as spatial buffers. They uptake K+ at sites of neuronal activity (mainly synapses) and release it at distant contacts with blood vessels. K+ channels belong to the largest super-family of ion channels, many of which are Ca2+-activated3. Based on their single channel conductance, they are divided into three subfamilies: big conductance (BK), intermediate conductance and small conductance. Astrocytic BK channels appear to be associated with voltage-gated and non-voltage-gated Ca2+ channels4-6. Regulation of intracellular Ca2+ homeostasis involves both entry from the extracellular space and release from intracellular sources (i.e.; endoplasmic reticulum and mitochondria). Astrocytic K+ and Ca2+ are involved in many fundamental pathophysiological functions, such as mitochondria biogenesis, cell survival, apoptosis, vascular tone, neurotransmitter release, and gene expression in the CNS3, 7-13. After CNS injury, accumulation of ions and water in astrocytes can lead to severe brain swelling.

Besides ion homeostasis, astrocytes play a key role in synapse formation and elimination11. Synapses underlie brain function. Three key processes drive synapse formation: first, axons choose from many potential postsynaptic partner cells. Second, after cell-cell contacts have formed, the portion of the axon that contacts the target cell differentiates into a presynaptic nerve terminal, while the domain of the target cell contacted by the axon differentiates into a specialized postsynaptic structure. Once formed, synapses continue to mature and rearrange. One striking aspect of later development is the elimination of a large proportion of synapses, a process that is usually accompanied by the growth and strengthening of surviving synapses. Synapse elimination plays a key role in refining initial patterns of brain connectivity14. The formation and elimination of synapses involves astrocytes. Neurons form few synapses when cultured in isolation, but many more when cultured in the presence of glia15. Astrocytes-derived molecules, such as a large matrix protein, thrombospondin (TSP), and cholesterol may modulate synapse function16. In addition, astrocytes may not only nurture synapses but also contribute proteoglycans that limit the ability of axons to reach out to new targets. Like microglia, astrocytes also have phagocytic functions, eliminating live synapses and clearing synaptic debris17-18.

Heme oxygenase (HO)-mediated antioxidant effects of astrocytes have been reported following CNS injury19-20. HO-2, a constitutively expressed form of the enzyme, regulates physiological cell function. HO-1, an inducible form of the enzyme, can be transcriptionally regulated by Nrf2 and biliverdin reductase (BVR). The Nrf2/HO-1 and/or BVR/HO-1 pathways produce carbon monoxide (CO), bilirubin (BR), and iron (Fe2+) by catabolizing heme. Among them, CO and BR can induce astrocytic mitochondrial biogenesis via activation of angiogenic and metabolic factors, such as AMP activated protein kinase a (AMPKa), hypoxia-inducible factor-1a (HIF-1a), estrogen-related receptor a (ERRa), and peroxisome proliferators-activated receptor γ-coactivator-1α (PGC-1α)21-22. Following CNS injury, astrocytes communicate with surrounding neuronal and vascular systems, leading to the clearance of disease-specific protein aggregates such as b-amyloid and a-synuclein18, 23-24. The HO-1/CO pathway in astrocytes suppresses protein aggregate-mediated toxicity19. However, persistent expression of HO-1 in astrocytes is deleterious, as excessive accumulation of iron can lead to inflammation and cell death25. HO metabolites are involved in astrocytic ion regulation, mitochondria biogenesis, and antioxidant effects.

Healthy astrocytic mitochondria may be essential for neuronal and vascular protection, as their inhibition is associated with neuronal cell death26. Further, the entry of astrocytic mitochondria into adjacent neurons can lead to amplified neuronal cell survival signals26. In addition, retinal ganglion cell axons release and transfer damaged mitochondria to astrocytes for disposal and recycling27. Therefore, Bioenergetic and antioxidant defenses in the brain and retina are coupled between neurons and astrocytes. Oxidative and nitrosative stress underlie the pathogenesis of a broad range of human diseases, in particular neurodegenerative disorders such as Alzheimer’s disease (AD), Parkinson’s disease (PD), and stroke. Within the brain, neurons are vulnerable to excess reactive oxygen/nitrogen species (ROS/RNS); therefore, their survival relies in part on antioxidant protection promoted by astrocytes19, 23, 28. This process is partly regulated by a key glycolysis-promoting enzyme (i.e.; 6-phosphofructo-2-kinase/fructose-2,6-bisphosphatase isoform 3), and is dependent on an appropriate supply of energy from astrocytes28. In this review, we will discuss the cellular and molecular mechanisms of astrocytes function in the healthy and diseased brains. Therefore, Antioxidant functions of astrocytes represent promising targets for therapeutic approaches to treat CNS injury such as AD, PD, and stroke.

Comment #2: line 66: what key glycolysis promoting enzyme?

Response to comment #2: We have included the name of the key glycolysis promoting enzyme, i.e., 6-phosphofructo-2-kinase/fructose-2,6-bisphosphatase isoform 3 (PFKFB3) in line 69-70, and additionally, provided the reference.

(Lines 69-71) This process is partly regulated by a key glycolysis-promoting enzyme (i.e.; 6-phosphofructo-2-kinase/fructose-2,6-bisphosphatase isoform 3), and is dependent on an appropriate supply of energy from astrocytes28.

Comment #3: lines 81-85: There is no logical connection between the second sentence and the rest of the paragraph.

Response to comment #3: We have revised the paragraph to improve the logical connection between the texts, and the changes are indicated in red ink.

(Lines 51-74) Heme oxygenase (HO)-mediated antioxidant effects of astrocytes have been reported following CNS injury19-20. HO-2, a constitutively expressed form of the enzyme, regulates physiological cell function. HO-1, an inducible form of the enzyme, can be transcriptionally regulated by nuclear factor erythroid 2-related factor 2 (Nrf2) and biliverdin reductase (BVR). The Nrf2/HO-1 and/or BVR/HO-1 pathways produce carbon monoxide (CO), bilirubin (BR), and iron (Fe2+) by catabolizing heme. Among them, CO and BR can induce astrocytic mitochondrial biogenesis via activation of angiogenic and metabolic factors, such as AMP activated protein kinase a (AMPKa), hypoxia-inducible factor-1a (HIF-1a), estrogen-related receptor a (ERRa), and peroxisome proliferators-activated receptor γ-coactivator-1α (PGC-1α)21-22. Following CNS injury, astrocytes communicate with surrounding neuronal and vascular systems, leading to the clearance of disease-specific protein aggregates such as b-amyloid and a-synuclein18, 23-24. The HO-1/CO pathway in astrocytes suppresses protein aggregate-mediated toxicity19. HO metabolites are involved in astrocytic ion regulation, mitochondria biogenesis, and antioxidant effects.

Bioenergetic and antioxidant defenses in the brain are coupled between neurons and astrocytes. Oxidative and nitrosative stress underlie the pathogenesis of a broad range of human diseases, in particular neurodegenerative disorders such as Alzheimer’s disease (AD), Parkinson’s disease (PD), and stroke. Within the brain, neurons are vulnerable to excess reactive oxygen/nitrogen species (ROS/RNS); therefore, their survival relies in part on antioxidant protection promoted by astrocytes19, 23, 28. This process is partly controlled by a key glycolysis-promoting enzyme (i.e.; 6-phosphofructo-2-kinase/fructose-2,6-bisphosphatase isoform 3) and is dependent on an appropriate supply of energy from astrocytes28. In this review, we discuss the cellular and molecular mechanisms of astrocyte function in the healthy and diseased brains. The antioxidant function of astrocytes present promising targets for therapeutic intervention for treating CNS injuries and diseases such as stroke, AD, and PD.

Comment #4: lines 87-95: References are needed.

Response to comment #4: We have included the relevant references. Additionally, as per Reviewer #3’s suggestion, we have provided a brief description on the role of P2X7 receptor for Ca2+ waves.

(Lines 76-86) Astrocytes sense when neurons are active, by depolarizing in response to neuronal K+ release, and by expressing neurotransmitter receptors similar to those of neurons. For example, Bergmann glia in the cerebellum express glutamate receptors29. Thus, glutamate released at cerebellar synapses affects not only postsynaptic neurons but also nearby astrocytes. The binding of these ligands to glial receptors increases the intracellular Ca2+ concentration. The astrocytes connect to neighboring astrocytes through gap junctions, allowing transfer of ions and small molecules between cells30. Pannexin hemichannel, a vertebrate homolog of the invertebrate innexin gap junction proteins, expels adenosine triphosphate (ATP) into the extracellular space31-32. Ca2+ waves are mediated largely by the binding of ATP to the P2X7 receptor33-34. This spread of Ca2+ through the astrocyte network occur over hundreds of micrometers30. It is likely that this Ca2+ wave modulates nearby neuronal activity by triggering the release of nutrients and regulating blood flow.

Comment #5: lines 131-139: There is no logical connection between the first sentence and the rest of the paragraph.

Response to comment #5: We have revised the paragraph to improve the logical flow of the text.

(Lines 122-130) Astrocyte hypertrophy is characterized by astrocyte swelling, enlargement and morphological changes and is key phenomenon in neuroinflammatory diseases35. The endotoxin lipopolysaccharide (LPS), high extracellular K+, glutamate, and ATP promote astrocyte hypertrophy in cultured astrocytes36. Primary cultures of mouse cortical astrocytes from newborn C57BL/6 mice show high expression of the a subunits of the L-type (Cav1.2 and Cav1.3) and P/Q-type (Cav2.1) VGCC under normal physiological conditions36. Furthermore, LPS increases Cav1.2 expression during astrocyte hypertrophy. Astrocytes isolated from mouse cortex transfected with siRNAs for Cav1.2 do not demonstrate astrocyte activation (e.g.; GFAP, nestin, and vimentin expression) and proliferation (e.g.; Ki67 expression) normally induced by LPS36.

Comment #6: line 156: COX2 is not a cytokine.

Response to comment #6: We have deleted COX2.

(Line 147) Inflammatory cytokines such as tumor necrosis factor a (TNFa) COX2 and prostaglandin E2 (PGE2)

Comment #7: lines 156-162: The first and second sentences are not well connected nor are they at all connected to the last two sentences.

Response to comment #7: We have revised the paragraph to improve the logical connection between sentences.

(Line 147-153) Inflammatory cytokines such as tumor necrosis factor a (TNFa) and prostaglandin E2 (PGE2) can be released from reactive astrocytes via kappa-light-chain-enhancer of activated B cells (NF-κB) signaling37-38. Reactive hippocampal astrocytes from mice, subjected to pilocarpine-induced status epilepticus, show upregulation of Cav1.3 or Cav2.1 expression39. Ischemic injury in the rat brain was reported to increase the expression of Cav1.2 in the reactive astrocytes located in the hippocampus and the cerebral cortex40. Following CNS injury, in conjunction with powerful, sequential cytosolic Ca2+ waves, the astrocytes promote TNFa, PGE2, ATP, glutamate, and D-serine secretion41.

Comment #8: line 165: Define BK channels.

Response to comment #8: BK channels have been previously defined in the Introduction Section.

(Lines 32-37) K+ channels belong to the largest super-family of ion channels, many of which are Ca2+-activated3. Based on their single channel conductance, they are divided into three subfamilies: big conductance (BK), intermediate conductance and small conductance. Astrocytic BK channels appear to be associated with voltage-gated and non-voltage-gated Ca2+ channels4-6.

Comment #9: lines 194-195: Earlier many of the same compounds described here as neurotransmitters were described as toxic cytokines. Which is it?

Response to comment #9: We have deleted PGE2 and TNF-a in this sentence. We thank you for your comment.

(Lines 196-197) Astrocytes upregulate and release many neurotransmitters such as PGE2, glutamate, TNF-a, ATP, and D-serine42,

Comment #10: lines 200-212: What does TSP mean? It is not clear what role TSPs have in the tripartite synapse.

Response to comment #10: TSP has been previously mentioned in the Introduction Section (Line 47-48). To demonstrate the role of TSPs in the tripartite synapse, we have included it in Line 207-210.

(Lines 47-48) Astrocytes-derived molecules, such as a large matrix protein, thrombospondin (TSP), and cholesterol may modulate synapse function16.

(Lines 207-210) TSPs are secreted multidomain glycoproteins found throughout the body including the CNS. TSP-induced synapse formation is ultrastructurally normal and shows proper alignment of the pre- and post-synaptic specializations in the tripartite synpase16. TSPs elicit functional presynaptic activity such as cycling of the synaptic vesicles16. Removal of TSPs from astrocyte-conditioned media diminishes synaptogenic activity in retinal ganglion cell cultures16.

Comment #11: lines 222-235: This paragraph describes two different processes that appear to have no clear relationship. Why are they combined?

Response to comment #11: We have deleted lines 222-225 for clarity.

(Original Lines 222-225) The crosstalk between astrocytes and neurons affects synapse elimination. Ganglion cell axons transfer mitochondria to astrocytes, leading to lysosomal degradation in astrocytes27. This process accounts for the majority of mitochondrial disposal and recycling, an energetically efficient pathway. The system involves phagocytic astrocyte activity on the optic nerve head43.

Comment #12: lines 244-245: what are A1 and A2 type astrocytes?

Response to comment #12: Reviewer #1 also had the same query. We have described the A1 and A2 astrocytes in line 185-195. A1 and A2 astrocyte are types of “reactive astrocytes” converted from diverse astrocytes, and were so termed by Liddelow S. A. et al.44. We moved the original description and added few more lines, 185-195.

(Lines 185-196) Astrocytes become reactive in response to CNS injury. Reactive astrocytes demonstrate heterogeneity, enabling their distinction into two types (i.e., A1 and A2 reactive astrocyte), and show reversible alterations in gene expression, cell hypertrophy, and formation of a glial scar45. The NF-κB pathway is closely involved in the deleterious effects of A1 astrocytes on the neurovascular unit1. Contrarily, the scar-forming A2 reactive astrocytes are beneficial as they encapsulate the injury or seal the damaged BBB by forming a glial scar44. Under neuroinflammatory conditions, the A1 type astrocytes show increased expression of genes that are destructive to synapses, whereas, in ischemic conditions, the A2 type astrocytes promote the expression of the genes beneficial to neuronal survival and growth1, 44. Activated microglia induce A1 type astrocytes by inducing IL-1a, TNFa, and C1q, and A1 astrocyte counts are elevated in AD and other neurodegenerative disorders44. In neuroinflammatory diseases, A1 astrocytes lose their ability to promote astrocyte-neuron connections and gain neurotoxicity.

(Original Lines 243-249) Under neuroinflammatory conditions, A1 type astrocytes show enhanced expression of genes that are destructive to synapses, while in ischemic conditions A2 type astrocytes promote expression of genes that are beneficial to neuronal survival and growth1, 44. Activated microglia induce A1 type of astrocytes by secretion of IL-1a, TNFa, and C1q, and A1 astrocytes are upregulated in AD and other neurodegenerative disorders44. In neuroinflammatory diseases, A1 astrocytes lose their ability to promote astrocyte-neuron connections and gain neurotoxicity.

(Original Lines 354-359) Reactive astrocytes have dual role in axonal death and regeneration. The NF-κB pathway is closely related to the deleterious A1 astrocyte-mediated effects on the neurovascular unit1. TNFa-mediated induction of the NF-κB pathway stimulates the increase of the proinflammatory enzyme, cyclooxygenase-2, and additionally, increases tau hyperphosphorylation via c-Jun N-terminal kinase (JNK)46. Scar-forming A2 reactive astrocytes are beneficial as they encapsulate injury or seal off the damaged BBB by forming a glial scar. The JAK-STAT3 pathway is involved in A2 astrocytes activation44. A2 reactive astrocytes are beneficial as they promote neurotrophic factors partly through the Janus kinase-Signal transducer and activator of transcription protein 3 JAK-STAT3 pathway44.

Comment #13: lines 269-271: tau is not a low molecular weight protein (~56 kD) and a tauopathy is a disease caused by aggregation of tau. The aggregates are called neurofibrillary tangles not tauopathies.

Response to comment #13: We have revised it as per the Reviewer’s comment. We thank you for your suggestion.

(Lines 262-266) Tau is a microtubule-associated protein (MAP) expressed in axons. Modified Tau proteins undergoing abnormal polymerization form the characteristic lesion called neurofibrillary tangle47. Progressive neurodegenerative disorders with pathological findings of filamentous inclusion bodies composed of MAP tau are referred to as tauopathies, and moreover, tau accumulation in astrocytes has been investigated48.

Comment #14: lines 277-284: Not clear how this treatment is beneficial or directly relevant to astrocytes.

Response to comment #14: S100b is mainly expressed in the astrocytes. Astrocyte-derived S100b exacerbates ROS/RNS production through the NF-κB pathway. We have included it in lines 276-282.

(Lines 277-283) Inhibition of IL-1R in 3xTg-AD mice by intraperitoneal injection of 200 mg of IL-1R antibody reduced the astrocyte-derived Ca2+ binding protein, S100b49. Astrocyte-derived S100b can upregulate iNOS in cultured rat astrocytes through NF-kB activation50, exacerbating oxidative/nitrosative stress in an autocrine manner. Astrocytic S100b secretion leads to tau hyperphosphorylation by inducing GSK-3b phosphorylation-mediated b-catenin degradation in human neural stem cells in an paracrine manner51. Therefore, healthy astrocytes contribute to proper neuronal functions and reactive inflammatory astrocytes accelerate neuronal pathology.

Comment #15: section 3.3: Much of this is discussed elsewhere. No need to repeat.

Response to comment #15: To avoid repetition, we have deleted some sentences as indicated below.

(Original Lines 56-59) Healthy astrocytic mitochondria may be essential for neuronal and vascular protection, as their inhibition is associated with neuronal cell death26. Further, the entry of astrocytic mitochondria into adjacent neurons can lead to amplified neuronal cell survival signals26. In addition, retinal ganglion cell axons release and transfer damaged mitochondria to astrocytes for disposal and recycling27.

(Original Lines 222-225) The crosstalk between astrocytes and neurons affects synapse elimination. Ganglion cell axons transfer mitochondria to astrocytes, leading to lysosomal degradation in astrocytes27. This process accounts for the majority of mitochondrial disposal and recycling, an energetically efficient pathway. The system involves phagocytic astrocyte activity on the optic nerve head43.

Comment #16: section 3.4: Much of this is discussed earlier. Should consolidate and not repeat.

Response to comment #16: To avoid redundancy, we have deleted few sentences in Section 3.4.

(Original Lines 336-342) In cultured human astrocytes, pretreatment and recovery of CO-releasing molecule-2 (CORM-2) can induce HO-1 expression in chronic conditions9. HO metabolites such as CO and BR regulate HIF-1a activity through activation of L-type Ca2+ channel in astrocytes22. In one study, HO-1-AMPKa-PGC-1a axis was shown to stabilize HIF-1a by reducing proline hydroxylase 2 (PHD2)-mediated proteasomal degradation of HIF-1a in a O2-dependent and ROS-independent manner22. In addition, CO and BR increase mitochondria biogenic factors such as PGC-1a, ERRa, HIF-1a, and VEGF in a mouse model of ischemic stroke9, 21, 52, possibly promoting astrocytic metabolic responses.

Comment #17: lines 347-350: Relevance of these sentences to the rest of the paragraph is not clear nor is the relevance to the review in general.

Response to comment #17: We thank you for your comment. We have deleted the following sentences to improve the clarity and relevance of this review.

(Original Lines 347-350) A recent study revealed that the laser ablation on astrocytic endfeet located on blood vessels did not result in leakage of Evans blue or dextran-conjugated fluorescein isothiocyanate from the stripped surface of the vessels53. Ablated astrocytes appear to be non-reactive, and also recovered blood vessels after stripped by laser ablation53.

Comment #18: lines 443-452: Not clear why this is relevant to astrocytes.

Response to comment #18: We have deleted one sentence, however, retained the rest. Here, we have tried to explain the role of BK channel in stroke. However, we could not find any direct evidence on the role of astrocyte BK channel in stroke. Instead, we found that a BK channel agonist for stroke patients failed in phase III of the clinical trial. In this review, we suggest that further examination is required to decipher the identity of the specific cell-type (i.e., astrocyte, neuron) that benefit from the activation of BK channels, subsequently conferring neuroprotection. Therefore, we have included (i.e., astrocyte, neuron).

(Original Lines 443-452) Cerebral ischemia/reperfusion injury is followed by a delayed secondary pathology including excitotoxic and inflammatory responses. Mitochondrial BK channels located on the inner membrane are known to play a critical protective role in ischemia/reperfusion injury54. BKa KO mice display larger infarct volume, more severe neurological scores, and higher mortality than their WT littermates following ischemia/reperfusion injury55. Administration of a BK channel opener (BMS-204352) was intravenously injected after MCAO in rats, and showed reduced cortical infarct size5. Despite promising preclinical results, the therapeutic efficacy of BMS-204352 failed to demonstrate improvement in a phase III clinical trial involving acute stroke patients (Reviewed in 3). Further examination is required to decipher the identity of the specific cell-type (i.e., astrocyte, neuron) that benefit from the activation of BK channels, subsequently conferring neuroprotection.

Comment #19: lines 475-476: Why would there be a difference? Why would induction of HO-1 by chemicals provide better buffering than HO-1 overexpression? The authors need to explain.

Response to comment #19: We have explained it in lines 459-464.

(Lines 459-464) Persistent expression of HO-1 in astrocytes is deleterious as excessive accumulation of iron can lead to inflammation and cell death25. HO-1 inducer transiently upregulates HO-1, concomitantly with enhanced levels of antioxidant proteins such as Nrf2 or BVR56-57, as well as mitochondrial ferritin58. More investigation is needed to elucidate the underlying mechanisms involved. It is possible that HO-1 inducers in astrocytes have more efficient iron buffering systems and antioxidant effects than HO-1 overexpression in astrocytes.

Comment #20: lines 478-486: Much of this was covered earlier. No need to repeat.

Response to comment #20: Since this paragraph was not repeated earlier, we would like to retain them.

(Original Lines 478-486) Astrocytes-derived CO production has been reported to contribute to vasodilation59, leading to the supply of O2 and nutrients to neighboring cells. Adenosine diphosphate (ADP) and NO are important signaling molecules in the brain, and both ADP and NO donors increases pial arteriolar diameter60. Dilation in response to ADP and ADP-dependent CO production were blocked by the metal porphyrin inhibitor of HO in astrocytes and cerebral microvessels60. CO and NO can activate BK channels in endothelial cells61. In addition, astrocytic-derived CO activates BK channels in smooth muscle cells directly, as well as via a NO-dependent pathway62. Therefore, astrocyte-derived CO can diffuse into endothelial and smooth muscle cells, leading to BK channel activation and consequent vasodilation.

Comment #21: lines 532-538: The authors need to explain what was being tested here. Clinical trials of what?

Response to comment #21: Vitamin E (antioxidant) + memantine (AD drug), or Vitamin E (antioxidant) + Selenium (AD drug) had been tested for their preventive and therapeutic effects in AD. However, they failed in the phase III of the clinical trials. We have described them in the original lines 532-538.

(Original Lines 532-538) Antioxidants have been applied in clinical trials because they are neuroprotective in animal models. Injection of oxidative/nitrosative stressed mice with phenyl-a-tert-butyl nitrone, a ROS/RNS scavenger, did not result in significant improvement of behavioral function63. Phase III clinical trials testing a combination of an antioxidant with a drug (i.e.; vitamin E plus memantine or selenium) for prevention or treatment of AD have been failed (reviewed in 64). Despite negative outcomes in large clinical trials, antioxidant agents are still important factors that may influence the critical balance between production and elimination of ROS/RNS.

References

1.              Liddelow, S. A.; Barres, B. A., Reactive Astrocytes: Production, Function, and Therapeutic Potential. Immunity 2017, 46 (6), 957-967.

2.              Abbott, N. J.; Ronnback, L.; Hansson, E., Astrocyte-endothelial interactions at the blood-brain barrier. Nature reviews. Neuroscience 2006, 7 (1), 41-53.

3.              Contet, C.; Goulding, S. P.; Kuljis, D. A.; Barth, A. L., BK Channels in the Central Nervous System. Int Rev Neurobiol 2016, 128, 281-342.

4.             Gueguinou, M.; Chantome, A.; Fromont, G.; Bougnoux, P.; Vandier, C.; Potier-Cartereau, M., KCa and Ca(2+) channels: the complex thought. Biochim Biophys Acta 2014, 1843 (10), 2322-33.

5.              Gribkoff, V. K.; Starrett, J. E., Jr.; Dworetzky, S. I., Maxi-K potassium channels: form, function, and modulation of a class of endogenous regulators of intracellular calcium. Neuroscientist 2001, 7 (2), 166-77.

6.              Price, D. L.; Ludwig, J. W.; Mi, H.; Schwarz, T. L.; Ellisman, M. H., Distribution of rSlo Ca2+-activated K+ channels in rat astrocyte perivascular endfeet. Brain Res 2002, 956 (2), 183-93.

7.              Seifert, G.; Henneberger, C.; Steinhauser, C., Diversity of astrocyte potassium channels: An update. Brain Res Bull 2018, 136, 26-36.

8.              Zhang, F. X.; Gadotti, V. M.; Souza, I. A.; Chen, L.; Zamponi, G. W., BK Potassium Channels Suppress Cavalpha2delta Subunit Function to Reduce Inflammatory and Neuropathic Pain. Cell Rep 2018, 22 (8), 1956-1964.

9.              Choi, Y. K.; Kim, J. H.; Lee, D. K.; Lee, K. S.; Won, M. H.; Jeoung, D.; Lee, H.; Ha, K. S.; Kwon, Y. G.; Kim, Y. M., Carbon Monoxide Potentiation of L-Type Ca2+ Channel Activity Increases HIF-1alpha-Independent VEGF Expression via an AMPKalpha/SIRT1-Mediated PGC-1alpha/ERRalpha Axis. Antioxid Redox Signal 2017, 27 (1), 21-36.

10.           Girouard, H.; Bonev, A. D.; Hannah, R. M.; Meredith, A.; Aldrich, R. W.; Nelson, M. T., Astrocytic endfoot Ca2+ and BK channels determine both arteriolar dilation and constriction. Proc Natl Acad Sci U S A 2010, 107 (8), 3811-6.

11.           Eroglu, C.; Barres, B. A., Regulation of synaptic connectivity by glia. Nature 2010, 468 (7321), 223-31.

12.           Agulhon, C.; Sun, M. Y.; Murphy, T.; Myers, T.; Lauderdale, K.; Fiacco, T. A., Calcium Signaling and Gliotransmission in Normal vs. Reactive Astrocytes. Front Pharmacol 2012, 3, 139.

13.           Zaichick, S. V.; McGrath, K. M.; Caraveo, G., The role of Ca(2+) signaling in Parkinson's disease. Dis Model Mech 2017, 10 (5), 519-535.

14.           Chung, W. S.; Clarke, L. E.; Wang, G. X.; Stafford, B. K.; Sher, A.; Chakraborty, C.; Joung, J.; Foo, L. C.; Thompson, A.; Chen, C.; Smith, S. J.; Barres, B. A., Astrocytes mediate synapse elimination through MEGF10 and MERTK pathways. Nature 2013, 504 (7480), 394-400.

15.           Barres, B. A., The mystery and magic of glia: a perspective on their roles in health and disease. Neuron 2008, 60 (3), 430-40.

16.           Christopherson, K. S.; Ullian, E. M.; Stokes, C. C.; Mullowney, C. E.; Hell, J. W.; Agah, A.; Lawler, J.; Mosher, D. F.; Bornstein, P.; Barres, B. A., Thrombospondins are astrocyte-secreted proteins that promote CNS synaptogenesis. Cell 2005, 120 (3), 421-33.

17.           Jung, Y. J.; Chung, W. S., Phagocytic Roles of Glial Cells in Healthy and Diseased Brains. Biomol Ther (Seoul) 2018, 26 (4), 350-357.

18.           Gomez-Arboledas, A.; Davila, J. C.; Sanchez-Mejias, E.; Navarro, V.; Nunez-Diaz, C.; Sanchez-Varo, R.; Sanchez-Mico, M. V.; Trujillo-Estrada, L.; Fernandez-Valenzuela, J. J.; Vizuete, M.; Comella, J. X.; Galea, E.; Vitorica, J.; Gutierrez, A., Phagocytic clearance of presynaptic dystrophies by reactive astrocytes in Alzheimer's disease. Glia 2018, 66 (3), 637-653.

19.           Hettiarachchi, N. T.; Boyle, J. P.; Dallas, M. L.; Al-Owais, M. M.; Scragg, J. L.; Peers, C., Heme oxygenase-1 derived carbon monoxide suppresses Abeta1-42 toxicity in astrocytes. Cell Death Dis 2017, 8 (6), e2884.

20.           Chen-Roetling, J.; Benvenisti-Zarom, L.; Regan, R. F., Cultured astrocytes from heme oxygenase-1 knockout mice are more vulnerable to heme-mediated oxidative injury. J Neurosci Res 2005, 82 (6), 802-10.

21.           Choi, Y. K.; Park, J. H.; Baek, Y. Y.; Won, M. H.; Jeoung, D.; Lee, H.; Ha, K. S.; Kwon, Y. G.; Kim, Y. M., Carbon monoxide stimulates astrocytic mitochondrial biogenesis via L-type Ca2+ channel-mediated PGC-1alpha/ERRalpha activation. Biochem Biophys Res Commun 2016, 479 (2), 297-304.

22.           Choi, Y. K.; Park, J. H.; Yun, J. A.; Cha, J. H.; Kim, Y.; Won, M. H.; Kim, K. W.; Ha, K. S.; Kwon, Y. G.; Kim, Y. M., Heme oxygenase metabolites improve astrocytic mitochondrial function via a Ca2+-dependent HIF-1alpha/ERRalpha circuit. PLoS One 2018, 13 (8), e0202039.

23.           Gonzalez-Reyes, R. E.; Nava-Mesa, M. O.; Vargas-Sanchez, K.; Ariza-Salamanca, D.; Mora-Munoz, L., Involvement of Astrocytes in Alzheimer's Disease from a Neuroinflammatory and Oxidative Stress Perspective. Front Mol Neurosci 2017, 10, 427.

24.           Lee, H. J.; Suk, J. E.; Patrick, C.; Bae, E. J.; Cho, J. H.; Rho, S.; Hwang, D.; Masliah, E.; Lee, S. J., Direct transfer of alpha-synuclein from neuron to astroglia causes inflammatory responses in synucleinopathies. J Biol Chem 2010, 285 (12), 9262-72.

25.           Schipper, H. M.; Song, W.; Tavitian, A.; Cressatti, M., The sinister face of heme oxygenase-1 in brain aging and disease. Prog Neurobiol 2019, 172, 40-70.

26.           Hayakawa, K.; Esposito, E.; Wang, X.; Terasaki, Y.; Liu, Y.; Xing, C.; Ji, X.; Lo, E. H., Transfer of mitochondria from astrocytes to neurons after stroke. Nature 2016, 535 (7613), 551-5.

27.           Davis, C. H.; Kim, K. Y.; Bushong, E. A.; Mills, E. A.; Boassa, D.; Shih, T.; Kinebuchi, M.; Phan, S.; Zhou, Y.; Bihlmeyer, N. A.; Nguyen, J. V.; Jin, Y.; Ellisman, M. H.; Marsh-Armstrong, N., Transcellular degradation of axonal mitochondria. Proc Natl Acad Sci U S A 2014, 111 (26), 9633-8.

28.           Fernandez-Fernandez, S.; Almeida, A.; Bolanos, J. P., Antioxidant and bioenergetic coupling between neurons and astrocytes. Biochem J 2012, 443 (1), 3-11.

29.           Piet, R.; Jahr, C. E., Glutamatergic and purinergic receptor-mediated calcium transients in Bergmann glial cells. J Neurosci 2007, 27 (15), 4027-35.

30.           Cornell-Bell, A. H.; Finkbeiner, S. M.; Cooper, M. S.; Smith, S. J., Glutamate induces calcium waves in cultured astrocytes: long-range glial signaling. Science 1990, 247 (4941), 470-3.

31.           Beckel, J. M.; Argall, A. J.; Lim, J. C.; Xia, J.; Lu, W.; Coffey, E. E.; Macarak, E. J.; Shahidullah, M.; Delamere, N. A.; Zode, G. S.; Sheffield, V. C.; Shestopalov, V. I.; Laties, A. M.; Mitchell, C. H., Mechanosensitive release of adenosine 5'-triphosphate through pannexin channels and mechanosensitive upregulation of pannexin channels in optic nerve head astrocytes: a mechanism for purinergic involvement in chronic strain. Glia 2014, 62 (9), 1486-501.

32.           Baranova, A.; Ivanov, D.; Petrash, N.; Pestova, A.; Skoblov, M.; Kelmanson, I.; Shagin, D.; Nazarenko, S.; Geraymovych, E.; Litvin, O.; Tiunova, A.; Born, T. L.; Usman, N.; Staroverov, D.; Lukyanov, S.; Panchin, Y., The mammalian pannexin family is homologous to the invertebrate innexin gap junction proteins. Genomics 2004, 83 (4), 706-16.

33.           Rubini, P.; Pagel, G.; Mehri, S.; Marquardt, P.; Riedel, T.; Illes, P., Functional P2X7 receptors at cultured hippocampal astrocytes but not neurons. Naunyn Schmiedebergs Arch Pharmacol 2014, 387 (10), 943-54.

34.           de Rivero Vaccari, J. P.; Dietrich, W. D.; Keane, R. W., Activation and regulation of cellular inflammasomes: gaps in our knowledge for central nervous system injury. Journal of cerebral blood flow and metabolism : official journal of the International Society of Cerebral Blood Flow and Metabolism 2014, 34 (3), 369-75.

35.           Lee, H.; Choi, Y. K., Regenerative Effects of Heme Oxygenase Metabolites on Neuroinflammatory Diseases. Int J Mol Sci 2018, 20 (1).

36.           Cheli, V. T.; Santiago Gonzalez, D. A.; Smith, J.; Spreuer, V.; Murphy, G. G.; Paez, P. M., L-type voltage-operated calcium channels contribute to astrocyte activation In vitro. Glia 2016, 64 (8), 1396-415.

37.           Alvarez, S.; Blanco, A.; Fresno, M.; Munoz-Fernandez, M. A., Nuclear factor-kappaB activation regulates cyclooxygenase-2 induction in human astrocytes in response to CXCL12: role in neuronal toxicity. Journal of neurochemistry 2010, 113 (3), 772-83.

38.           Blanco, A.; Alvarez, S.; Fresno, M.; Munoz-Fernandez, M. A., Amyloid-beta induces cyclooxygenase-2 and PGE2 release in human astrocytes in NF-kappa B dependent manner. J Alzheimers Dis 2010, 22 (2), 493-505.

39.           Xu, J. H.; Long, L.; Tang, Y. C.; Hu, H. T.; Tang, F. R., Ca(v)1.2, Ca(v)1.3, and Ca(v)2.1 in the mouse hippocampus during and after pilocarpine-induced status epilepticus. Hippocampus 2007, 17 (3), 235-51.

40.           Chung, Y. H.; Shin, C. M.; Kim, M. J.; Cha, C. I., Enhanced expression of L-type Ca2+ channels in reactive astrocytes after ischemic injury in rats. Neurosci Lett 2001, 302 (2-3), 93-6.

41.           Wang, X.; Lou, N.; Xu, Q.; Tian, G. F.; Peng, W. G.; Han, X.; Kang, J.; Takano, T.; Nedergaard, M., Astrocytic Ca2+ signaling evoked by sensory stimulation in vivo. Nat Neurosci 2006, 9 (6), 816-23.

42.           Takano, T.; Oberheim, N.; Cotrina, M. L.; Nedergaard, M., Astrocytes and ischemic injury. Stroke; a journal of cerebral circulation 2009, 40 (3 Suppl), S8-12.

43.           Davis, C. H.; Marsh-Armstrong, N., Discovery and implications of transcellular mitophagy. Autophagy 2014, 10 (12), 2383-4.

44.           Liddelow, S. A.; Guttenplan, K. A.; Clarke, L. E.; Bennett, F. C.; Bohlen, C. J.; Schirmer, L.; Bennett, M. L.; Munch, A. E.; Chung, W. S.; Peterson, T. C.; Wilton, D. K.; Frouin, A.; Napier, B. A.; Panicker, N.; Kumar, M.; Buckwalter, M. S.; Rowitch, D. H.; Dawson, V. L.; Dawson, T. M.; Stevens, B.; Barres, B. A., Neurotoxic reactive astrocytes are induced by activated microglia. Nature 2017, 541 (7638), 481-487.

45.           Anderson, M. A.; Ao, Y.; Sofroniew, M. V., Heterogeneity of reactive astrocytes. Neurosci Lett 2014, 565, 23-9.

46.           Ploia, C.; Antoniou, X.; Sclip, A.; Grande, V.; Cardinetti, D.; Colombo, A.; Canu, N.; Benussi, L.; Ghidoni, R.; Forloni, G.; Borsello, T., JNK plays a key role in tau hyperphosphorylation in Alzheimer's disease models. J Alzheimers Dis 2011, 26 (2), 315-29.

47.           Wood, J. G.; Mirra, S. S.; Pollock, N. J.; Binder, L. I., Neurofibrillary tangles of Alzheimer disease share antigenic determinants with the axonal microtubule-associated protein tau (tau). Proc Natl Acad Sci U S A 1986, 83 (11), 4040-3.

48.           Leyns, C. E. G.; Holtzman, D. M., Glial contributions to neurodegeneration in tauopathies. Mol Neurodegener 2017, 12 (1), 50.

49.           Kitazawa, M.; Cheng, D.; Tsukamoto, M. R.; Koike, M. A.; Wes, P. D.; Vasilevko, V.; Cribbs, D. H.; LaFerla, F. M., Blocking IL-1 signaling rescues cognition, attenuates tau pathology, and restores neuronal beta-catenin pathway function in an Alzheimer's disease model. J Immunol 2011, 187 (12), 6539-49.

50.           Lam, A. G.; Koppal, T.; Akama, K. T.; Guo, L.; Craft, J. M.; Samy, B.; Schavocky, J. P.; Watterson, D. M.; Van Eldik, L. J., Mechanism of glial activation by S100B: involvement of the transcription factor NFkappaB. Neurobiol Aging 2001, 22 (5), 765-72.

51.           Esposito, G.; Scuderi, C.; Lu, J.; Savani, C.; De Filippis, D.; Iuvone, T.; Steardo, L., Jr.; Sheen, V.; Steardo, L., S100B induces tau protein hyperphosphorylation via Dickopff-1 up-regulation and disrupts the Wnt pathway in human neural stem cells. J Cell Mol Med 2008, 12 (3), 914-27.

52.           Choi, Y. K.; Kim, C. K.; Lee, H.; Jeoung, D.; Ha, K. S.; Kwon, Y. G.; Kim, K. W.; Kim, Y. M., Carbon monoxide promotes VEGF expression by increasing HIF-1alpha protein level via two distinct mechanisms, translational activation and stabilization of HIF-1alpha protein. J Biol Chem 2010, 285 (42), 32116-25.

53.           Kubotera, H.; Ikeshima-Kataoka, H.; Hatashita, Y.; Allegra Mascaro, A. L.; Pavone, F. S.; Inoue, T., Astrocytic endfeet re-cover blood vessels after removal by laser ablation. Sci Rep 2019, 9 (1), 1263.

54.           Singh, H.; Lu, R.; Bopassa, J. C.; Meredith, A. L.; Stefani, E.; Toro, L., MitoBK(Ca) is encoded by the Kcnma1 gene, and a splicing sequence defines its mitochondrial location. Proc Natl Acad Sci U S A 2013, 110 (26), 10836-41.

55.           Liao, Y.; Kristiansen, A. M.; Oksvold, C. P.; Tuvnes, F. A.; Gu, N.; Runden-Pran, E.; Ruth, P.; Sausbier, M.; Storm, J. F., Neuronal Ca2+-activated K+ channels limit brain infarction and promote survival. PLoS One 2010, 5 (12), e15601.

56.           Wang, B.; Cao, W.; Biswal, S.; Dore, S., Carbon monoxide-activated Nrf2 pathway leads to protection against permanent focal cerebral ischemia. Stroke; a journal of cerebral circulation 2011, 42 (9), 2605-10.

57.           Rochette, L.; Zeller, M.; Cottin, Y.; Vergely, C., Redox Functions of Heme Oxygenase-1 and Biliverdin Reductase in Diabetes. Trends Endocrinol Metab 2018, 29 (2), 74-85.

58.           Yu, X.; Song, N.; Guo, X.; Jiang, H.; Zhang, H.; Xie, J., Differences in vulnerability of neurons and astrocytes to heme oxygenase-1 modulation: Implications for mitochondrial ferritin. Sci Rep 2016, 6, 24200.

59.           Leffler, C. W.; Parfenova, H.; Fedinec, A. L.; Basuroy, S.; Tcheranova, D., Contributions of astrocytes and CO to pial arteriolar dilation to glutamate in newborn pigs. Am J Physiol Heart Circ Physiol 2006, 291 (6), H2897-904.

60.           Kanu, A.; Leffler, C. W., Roles of glia limitans astrocytes and carbon monoxide in adenosine diphosphate-induced pial arteriolar dilation in newborn pigs. Stroke; a journal of cerebral circulation 2009, 40 (3), 930-5.

61.           Kanu, A.; Leffler, C. W., Carbon monoxide and Ca2+-activated K+ channels in cerebral arteriolar responses to glutamate and hypoxia in newborn pigs. Am J Physiol Heart Circ Physiol 2007, 293 (5), H3193-200.

62.           Li, A.; Xi, Q.; Umstot, E. S.; Bellner, L.; Schwartzman, M. L.; Jaggar, J. H.; Leffler, C. W., Astrocyte-derived CO is a diffusible messenger that mediates glutamate-induced cerebral arteriolar dilation by activating smooth muscle Cell KCa channels. Circ Res 2008, 102 (2), 234-41.

63.           Choi, Y. K.; Maki, T.; Mandeville, E. T.; Koh, S. H.; Hayakawa, K.; Arai, K.; Kim, Y. M.; Whalen, M. J.; Xing, C.; Wang, X.; Kim, K. W.; Lo, E. H., Dual effects of carbon monoxide on pericytes and neurogenesis in traumatic brain injury. Nat Med 2016, 22 (11), 1335-1341.

64.           Chun, H.; Lee, C. J., Reactive astrocytes in Alzheimer's disease: A double-edged sword. Neurosci Res 2018, 126, 44-52.

Reviewer 3 Report

This review focuses on calcium and potassium channels in astrocytes and some specific antioxidant systems in astrocytes, and mitochondria- the title should be changed to reflect this restricted scope of the review. 

In this reviewer’s opinion, to many important roles of astrocytes in the CNS were mentioned in passing without very much detail (e.g. glucose transport and distribution, glutamate uptake a the synapse, astrocyte control of synapse formation, lipid and hormone processing) while large sections cover calcium and potassium channels. BK channels are discussed extensively while Kir channels are mentioned in passing.

If the title and abstract are changed to reflect the focus on BK channels and HO-1 then my opinion is to accept with minor revision.

If title is to be kept as is then my opinion is that this manuscript requires major revision to better review the complete current list of important roles for astrocytes in the CNS. 

If the review is intended to cover the Roles of Astrocytes in the CNS then a more complete list of roles should be discussed with greater depth.

Given the journal, system Xc-  and glutathione should be covered in this review.

Astrocyte role in glucose and lactate uptake/processing/distribution should be covered more thoroughly.

Line 93- Along with a component mediated by gap junctions, calcium waves are also mediated in large part by P2X7 receptors and pannexin paracellular signaling through ATP, which could be an important point for this review.

Line 117- I think “are followed” should be “are as follows”

Line 321- What doe it mean the AKAP12 is partly expressed in retinal astrocytes? Does this mean that part of the protein is expressed? Some astrocytes express it? This sentence does not seem to make sense as written.

Line 329- The reviewer understands that BBB breakdown can be separate from angiogenesis but this section may be confusing to readers. If additional potential mechanisms by which VEGF decreases BBB while stimulating angiogenesis can be briefly mentioned it might decrease reader confusion.

Line 350 – this sentence is confusing. I had to read it 3 or 4 times to understand that the authors meant that the astrocyte processes regenerated membrane to surround the vessel after the original astrocyte endfeet were ablated.

Line 415 – The word “Thus,” does not seem to be appropriate in this place.

Author Response

Response to Comments by Reviewer 3

This review focuses on calcium and potassium channels in astrocytes and some specific antioxidant systems in astrocytes, and mitochondria- the title should be changed to reflect this restricted scope of the review. 

In this reviewer’s opinion, to many important roles of astrocytes in the CNS were mentioned in passing without very much detail (e.g. glucose transport and distribution, glutamate uptake a the synapse, astrocyte control of synapse formation, lipid and hormone processing) while large sections cover calcium and potassium channels. BK channels are discussed extensively while Kir channels are mentioned in passing.

Comment #1: If the title and abstract are changed to reflect the focus on BK channels and HO-1 then my opinion is to accept with minor revision.

Response to comment #1: The title and the abstract has been changed according to your suggestion.

(Lines 2-23) The Role of Astrocytes in the Central Nervous System Focused on BK channel and Heme Oxygenase Metabolites: A Review

Abstract: Astrocytes outnumber neurons in the human brain, and they play a key role in numerous functions within the central nervous system (CNS), including glutamate, ion (i.e. Ca2+, K+) and water homeostasis, defense against oxidative/nitrosative stress, energy storage, mitochondria biogenesis, scar formation, tissue repair via angiogenesis and neurogenesis, and synapse modulation. After CNS injury, astrocytes communicate with surrounding neuronal and vascular systems, leading to clearance of disease-specific protein aggregates such as b-amyloid, and a-synuclein. Astrocytic big conductance K+ (BK) channel play a role in these processes. Recently, potential therapeutic agents that target astrocytes have been tested for their potential to repair the brain. In this review, we discuss the role of BK channel and antioxidant agents such as heme oxygenase metabolites following CNS injury. A better understanding of the cellular and molecular mechanisms of astrocytes functions in the healthy and diseased brains will greatly contribute to the development of therapeutic approaches following CNS injury such as Alzheimer’s disease, Parkinson’s disease, and stroke.

Key words: astrocytes; BK channel; oxidative/nitrosative stress; heme oxygenase metabolites

If title is to be kept as is then my opinion is that this manuscript requires major revision to better review the complete current list of important roles for astrocytes in the CNS.

If the review is intended to cover the Roles of Astrocytes in the CNS then a more complete list of roles should be discussed with greater depth.

Given the journal, system Xc-  and glutathione should be covered in this review.

Astrocyte role in glucose and lactate uptake/processing/distribution should be covered more thoroughly.

Comment #2: Line 93- Along with a component mediated by gap junctions, calcium waves are also mediated in large part by P2X7 receptors and pannexin paracellular signaling through ATP, which could be an important point for this review.

Response to comment #2: As per your suggestion, we have added it.

(Lines 80-86) The astrocytes connect to neighboring astrocytes through gap junctions, allowing transfer of ions and small molecules between cells1. Pannexin hemichannel, a vertebrate homolog of the invertebrate innexin gap junction proteins, expels adenosine triphosphate (ATP) into the extracellular space2-3. Ca2+ waves are mediated largely by the binding of ATP to the P2X7 receptor4-5. This spread of Ca2+ through the astrocyte network occur over hundreds of micrometers1. It is likely that this Ca2+ wave modulates nearby neuronal activity by triggering the release of nutrients and regulating blood flow.

Comment #3: Line 117- I think “are followed” should be “are as follows”

Response to comment #3: We have changed it as per your suggestion.

(Lines 106-109) VGCCs are divided into two groups: high-voltage-operated Ca2+ channels (L-, P/Q-, N- and R-types), and low-voltage-activated channel (T-type)6-8. a1-subunits of VGCCs are as follows: L-type (Cav1.1-Cav1.4), P/Q-type (Cav2.1), N-type (Cav2.2), R-type (Cav2.3), and T type (Cav3.1-3.3) Ca2+ channels9.

Comment #4: Line 321- What doe it mean the AKAP12 is partly expressed in retinal astrocytes? Does this mean that part of the protein is expressed? Some astrocytes express it? This sentence does not seem to make sense as written.

Response to comment #4: Part of the protein is expressed in the retinal astrocytes. We have changed it as per your suggestion.

(Lines 318-319) Part of A-kinase anchor protein 12 (AKAP12) protein is expressed in the retinal astrocytes, and can downregulate HIF-1a protein stability in a O2-dependent manner10.

Comment #5: Line 329- The reviewer understands that BBB breakdown can be separate from angiogenesis but this section may be confusing to readers. If additional potential mechanisms by which VEGF decreases BBB while stimulating angiogenesis can be briefly mentioned it might decrease reader confusion.

Response to comment #5: We have included the role of VEGF in the acute and chronic phase of CNS injury.

(Lines 326-329) A HIF-1a downstream factor, VEGF, also has protective effects against ischemic injury by stimulating angiogenesis, mitochondria biogenesis, neuronal survival, and neurogenesis11-14. Despite its deleterious effects on the BBB in the acute phase, VEGF is involved in regeneration in the chronic phase of CNS injury.

Comment #6: Line 350 – this sentence is confusing. I had to read it 3 or 4 times to understand that the authors meant that the astrocyte processes regenerated membrane to surround the vessel after the original astrocyte endfeet were ablated.

Response to comment #6: We thank you for your comment. We have deleted these sentences to improve the clarity and relevance of this review.

(Original Lines 347-350) A recent study revealed that the laser ablation on astrocytic endfeet located on blood vessels did not result in leakage of Evans blue or dextran-conjugated fluorescein isothiocyanate from the stripped surface of the vessels15. Ablated astrocytes appear to be non-reactive, and also recovered blood vessels after stripped by laser ablation15.

Comment #7: Line 415 – The word “Thus,” does not seem to be appropriate in this place.

Response to comment #7: As per your suggestion, we have deleted it it.

(Lines 400-402) A In physiological conditions, astrocytes also express a-synuclein at very low levels. Thus, a-synuclein may contribute to synaptic vesicle biogenesis and dynamics, and neurotransmission16.

References

1.              Cornell-Bell, A. H.; Finkbeiner, S. M.; Cooper, M. S.; Smith, S. J., Glutamate induces calcium waves in cultured astrocytes: long-range glial signaling. Science 1990, 247 (4941), 470-3.

2.              Beckel, J. M.; Argall, A. J.; Lim, J. C.; Xia, J.; Lu, W.; Coffey, E. E.; Macarak, E. J.; Shahidullah, M.; Delamere, N. A.; Zode, G. S.; Sheffield, V. C.; Shestopalov, V. I.; Laties, A. M.; Mitchell, C. H., Mechanosensitive release of adenosine 5'-triphosphate through pannexin channels and mechanosensitive upregulation of pannexin channels in optic nerve head astrocytes: a mechanism for purinergic involvement in chronic strain. Glia 2014, 62 (9), 1486-501.

3.              Baranova, A.; Ivanov, D.; Petrash, N.; Pestova, A.; Skoblov, M.; Kelmanson, I.; Shagin, D.; Nazarenko, S.; Geraymovych, E.; Litvin, O.; Tiunova, A.; Born, T. L.; Usman, N.; Staroverov, D.; Lukyanov, S.; Panchin, Y., The mammalian pannexin family is homologous to the invertebrate innexin gap junction proteins. Genomics 2004, 83 (4), 706-16.

4.              Rubini, P.; Pagel, G.; Mehri, S.; Marquardt, P.; Riedel, T.; Illes, P., Functional P2X7 receptors at cultured hippocampal astrocytes but not neurons. Naunyn Schmiedebergs Arch Pharmacol 2014, 387 (10), 943-54.

5.              de Rivero Vaccari, J. P.; Dietrich, W. D.; Keane, R. W., Activation and regulation of cellular inflammasomes: gaps in our knowledge for central nervous system injury. Journal of cerebral blood flow and metabolism : official journal of the International Society of Cerebral Blood Flow and Metabolism 2014, 34 (3), 369-75.

6.              Cheli, V. T.; Santiago Gonzalez, D. A.; Smith, J.; Spreuer, V.; Murphy, G. G.; Paez, P. M., L-type voltage-operated calcium channels contribute to astrocyte activation In vitro. Glia 2016, 64 (8), 1396-415.

7.              Barres, B. A.; Koroshetz, W. J.; Chun, L. L.; Corey, D. P., Ion channel expression by white matter glia: the type-1 astrocyte. Neuron 1990, 5 (4), 527-44.

8.              Puro, D. G.; Hwang, J. J.; Kwon, O. J.; Chin, H., Characterization of an L-type calcium channel expressed by human retinal Muller (glial) cells. Brain Res Mol Brain Res 1996, 37 (1-2), 41-8.

9.              Striessnig, J.; Ortner, N. J.; Pinggera, A., Pharmacology of L-type Calcium Channels: Novel Drugs for Old Targets? Curr Mol Pharmacol 2015, 8 (2), 110-22.

10.           Choi, Y. K.; Kim, J. H.; Kim, W. J.; Lee, H. Y.; Park, J. A.; Lee, S. W.; Yoon, D. K.; Kim, H. H.; Chung, H.; Yu, Y. S.; Kim, K. W., AKAP12 regulates human blood-retinal barrier formation by downregulation of hypoxia-inducible factor-1alpha. J Neurosci 2007, 27 (16), 4472-81.

11.           Arany, Z.; Foo, S. Y.; Ma, Y.; Ruas, J. L.; Bommi-Reddy, A.; Girnun, G.; Cooper, M.; Laznik, D.; Chinsomboon, J.; Rangwala, S. M.; Baek, K. H.; Rosenzweig, A.; Spiegelman, B. M., HIF-independent regulation of VEGF and angiogenesis by the transcriptional coactivator PGC-1alpha. Nature 2008, 451 (7181), 1008-12.

12.           Wright, G. L.; Maroulakou, I. G.; Eldridge, J.; Liby, T. L.; Sridharan, V.; Tsichlis, P. N.; Muise-Helmericks, R. C., VEGF stimulation of mitochondrial biogenesis: requirement of AKT3 kinase. FASEB J 2008, 22 (9), 3264-75.

13.           Wang, Y.; Jin, K.; Mao, X. O.; Xie, L.; Banwait, S.; Marti, H. H.; Greenberg, D. A., VEGF-overexpressing transgenic mice show enhanced post-ischemic neurogenesis and neuromigration. J Neurosci Res 2007, 85 (4), 740-7.

14.           Cao, L.; Jiao, X.; Zuzga, D. S.; Liu, Y.; Fong, D. M.; Young, D.; During, M. J., VEGF links hippocampal activity with neurogenesis, learning and memory. Nat Genet 2004, 36 (8), 827-35.

15.           Kubotera, H.; Ikeshima-Kataoka, H.; Hatashita, Y.; Allegra Mascaro, A. L.; Pavone, F. S.; Inoue, T., Astrocytic endfeet re-cover blood vessels after removal by laser ablation. Sci Rep 2019, 9 (1), 1263.

16.           Stefanis, L., alpha-Synuclein in Parkinson's disease. Cold Spring Harb Perspect Med 2012, 2 (2), a009399.

Round 2

Reviewer 2 Report

I still have a hard time with the organization of this review in that the different ideas and experiments are not well connected. However, as it seems that this is not going to be corrected, I will defer to the opinions of the other reviewers and the editor.